# Predicting individual differences of fear and cognitive learning and extinction

C. A. Gomes [1,2,3,4] ✉, D. R. Bach [5,6], A. Razi [5,7,8,9], G. Batsikadze [2,3,4], S. Elsenbruch [10], H. Engler [3,11], T. M. Ernst [2,3,4], M. C. Fellner[1], C. Fraenz[12], E. Genç[12], A. Klass[13], F. Labrenz [10], S. Lissek[13], C. J. Merz [14], D. Metzen[15], A. Nostadt [13,16], R. J. Pawlik[2,3], J. E. Schneider [1,17], M. Tegenthoff[13], A. Thieme [2,3], O. T. Wolf[14], O. Güntürkün [4,18,19], H. H. Quick[4,20], R. Kumsta[17], D. Timmann[2,3,4], T. Spisak [3] & N. Axmacher [1,4]

The ability to acquire new information and to modify previously learned knowledge are critical in an ever-changing world. However, the efficacy of learning is notably variable among individuals, with extinction learning being the epitome of such variability. Abundant studies have identified a core network of brain regions including the amygdala, hippocampus, dorsal anterior cingulate cortex (ACC), ventromedial prefrontal cortex (PFC) and, more recently, the cerebellum, as key players in learning and extinction. Yet, the precise interactions within this network and their relationship to individual learning abilities and extinction have remained largely unexplored. In the present study, we examined how functional (FC), effective (EC), and structural (SC) connectivity patterns in the core learning network allow the prediction of individual differences in the efficacy of learning, extinction, and renewal. Analysing a large dataset of over 500 participants across a multitude of paradigms, our results revealed that FC predicted better acquisition, with a central role of ACC and hippocampus, whereas SC, involving ACC and amygdala, predicted higher levels of extinction learning. EC results suggested a predominantly inhibitory coupling among core learning network nodes, with paradigm-specific EC connectivity patterns predicting learning. Our predictions not only generalised between fear and cognitive predictive learning paradigms but were also successful in predicting learning from task-related FC and simulated data. Together, these results describe the multimodal neural determinants of learning, extinction, and renewal, and may inform individualised interventions for affective disorders based on neural connectivity patterns.

The ability to learn from experience is a hallmark of every living system, from humans down to single-cell organisms. This ability differs strongly between individuals: While some are able to acquire new information quickly and display steep learning rates, others are much slower[1]. These differences concern abilities as widespread as the formation of new episodic memories, the development of novel practical skills, or the gradual learning about the putative outcome of actions.

Since our world is constantly changing, it is equally important to cease responding to previously memorised information once it is no

longer valid. This process is called extinction learning and involves the acquisition of two distinct memory traces. The first represents the initial association that is largely left intact, while the second is an association of inhibitory nature that suppresses the activation of the first trace[2]. These inhibited associative responses can return under diverse conditions and thus turn into invasive components of psychopathology[3]. The societal and clinical relevance of extinction learning and its associated problems can hardly be overestimated. According to Craske et al., more than 60 million European Union citizens suffer from anxiety disorders[4]. Importantly, extinction learning is itself strongly context-dependent, since presentation of a conditioned stimulus (CS) outside its extinction context tends to induce return of the conditioned response (CR), a phenomenon known as renewal[2].

The ability to extinguish previously acquired information and the propensity for renewal show pronounced individual differences, which may not only account for a person's ability to flexibly update knowledge but also their vulnerability or resilience to psychopathology, specifically regarding anxiety disorders[5]. It is likely that these differences reflect a combination of both stable (trait) and variable (state) measures. For example, the ability to acquire novel information and to update existing information, as well as the re-occurrence of extinguished memory traces, depend on age[6,7], sex[8,9], and personality traits such as trait anxiety and sensation seeking[10,11], but are also modulated by acute psychosocial stress and/or state anxiety[12,13]. Understanding the neural determinants of individual differences in learning, extinction, and renewal is thus not only a window into the mechanisms of extinction but may prove useful in our understanding of disorders that affect this ability and their potential treatments.

The brain structures involved in fear conditioning are relatively well-known. Animal and human research converge towards the idea that the amygdala (AMY) stores the associations between the CS and the unconditioned stimulus (US), whereas the hippocampus (HIP) encodes context information[14,15]. The dorsal anterior cingulate cortex (ACC) and ventromedial prefrontal cortex (PFC) have prominent roles in fear appraisal and safety learning, respectively[16]. More recently, it has also been proposed that the cerebellum (CEB) provides predictions of upcoming sensory events during associative tasks[17].

The mechanisms of extinction learning putatively differ from those supporting initial learning and may be more complex, since they require the formation of a second associative trace of an inhibitory nature. Moreover, despite the apparent ubiquity of learning and extinction in both fear conditioning and cognitive predictive learning contexts, it remains an open question whether these processes require the same or different neural determinants. For instance, whereas HIP and PFC support context-dependent extinction learning in both fear[18–20] and predictive learning[21–23] paradigms, the involvement and role of the AMY may be less universal than previously assumed[24,25]: Suppression of AMY activity by PFC enables extinction following aversive learning[14], whereas AMY activity increases during extinction in both appetitive[26] and predictive learning[21] tasks, presumably related to salience or novelty processing. Thus, although accumulated evidence points to the involvement of a similar set of brain regions in various conditioning-based learning paradigms, the macroscale network connectivity patterns that support these different kinds of learning remain unclear. Furthermore, there is a paucity of studies devoted to investigating individual differences in learning efficacy from inter-areal connectivity patterns.

Resting-state fMRI (rs-fMRI) has proved to be a reliable and convenient technique to measure intrinsic brain connectivity in large participant samples. Indeed, several studies have successfully predicted performance in various (non-)cognitive traits, as well as vulnerability or resilience to mental disorders[27–29], from brain connectivity patterns. Still, the extant studies examining brain-behaviour relationships using rs-fMRI have mostly focused on functional connectivity (FC). While this method appears to reflect inter-regional interactions

between local neural assemblies[30], it does not convey information about the direction of these interactions. By contrast, recent advances in effective connectivity (EC) now allow the characterization of causal interactions among brain areas at rest[31], which may provide important complementary information regarding the complex relationship between brain connectivity and individual differences in learning, extinction, and renewal. Even though correlation-based FC and EC are mathematically related, they differ fundamentally in that FC only accounts for linear, undirected statistical dependencies, whereas EC measures the directed causal influence that one brain region exerts over another[32,33].

In addition to functional and effective interactions, pronounced individual differences have been found in patterns of structural connectivity (SC) that reflect the integrity and effectivity of axonal information transfer. SC can be quantified using tractography, a technique based on diffusion-weighted imaging (DWI) that generates streamlines as a proxy for white matter fibre tracts across brain regions[34]. FC and SC are known to be related to some extent[35,36], but this relationship is complex. FC-SC correlations have been shown to depend on the specific network connections being examined[37], and the existence of strong FC in the absence of direct structural connections suggests that FC between two regions may rely on SC via a common third region[38].

The relationship between EC and SC is even less clear, although recent evidence suggests that constructing structurally-informed dynamic causal models (DCMs) of EC can outperform structurally-naïve DCMs by drastically improving group-level model evidence[39]. Nevertheless, it remains unclear how these two types of connectivity compare in terms of their ability to predict cognitive variables. In summary, cognition depends on a complex interplay between FC, EC, and SC, which has prompted researchers' calls for an integrative approach[40,41].

In the present study, we set out to investigate not only the neural connectivity patterns supporting learning and extinction but also whether these patterns generalise across different types of learning paradigms. As mentioned above, various forms of learning, such as fear learning and cognitive predictive learning, appear to rely on overlapping neural circuitry, as both involve acquiring and updating associative contingencies. While distinct paradigms may recruit additional regions based on task-specific demands (e.g., the piriform cortex in olfactory conditioning), a core learning network appears to be commonly engaged across different forms of associative learning. By comparing fear learning and cognitive predictive learning, our analysis aimed to identify this shared neural architecture and its role in learning and extinction. We analysed FC, EC and SC patterns within this network in a large multicenter dataset of over 500 individuals from a collaborative project involving different types of learning and extinction (see Fig. 1C, Supplemental text: Experimental paradigms and Fig. S5).

## Results

Because participants were scanned at three different locations using three different 3T MRI systems from two vendors, we carefully harmonised and optimised scanning sequences across centres. Specifically, we tested the stability of our resting-state and diffusion acquisition protocols by scanning two individuals over the course of three years on all three scanners using the exact imaging parameters of the actual studies. FC and SC measures demonstrated high test–retest reliability, both when using whole-brain region-of-interest (ROI) and when restricting the analyses to the study-specific ROIs (all Cronbach's alpha > .80; Figs. S1–2).

We then acquired rs-fMRI and DWI data from a large group of participants (rs-fMRI: N = 509; DWI: N = 463) who took part in either a fear learning (FL; studies S1, S2, S3, S5, and S6) or a cognitive predictive learning (PL; study S4) experiment (Fig. 1C; see also Methods and Supplemental Methods: Experimental paradigms for more details). The FL studies employed variations of classical fear conditioning, in

**A. Number of rs-fMRI and DWI datasets**

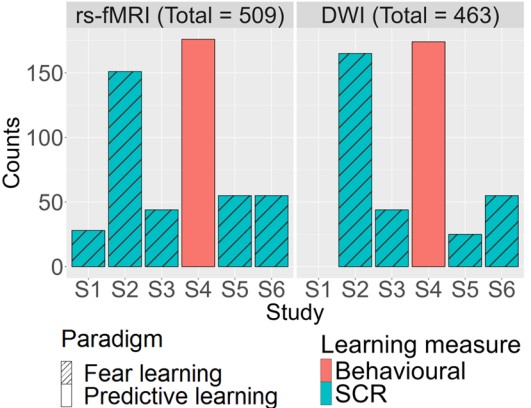

Paradigm
Fear learning
Predictive learning

Learning measure
Behavioural
SCR

**B. Regions of interest (ROIs) used in the present study**

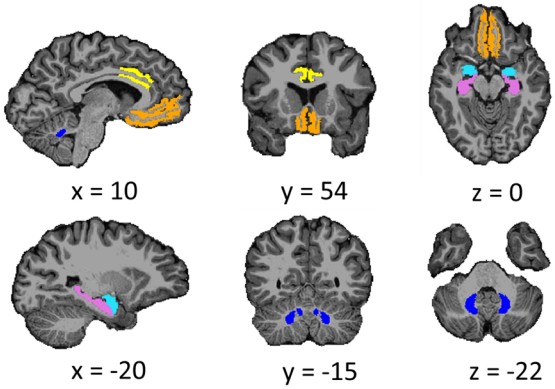

Ventral medial prefrontal cortex (PFC)
Dorsal anterior cingulate cortex (ACC)
Amygdala (AMY)
Hippocampus (HIP)
Cerebellar nuclei (CEB)

**C. Experimental paradigms**

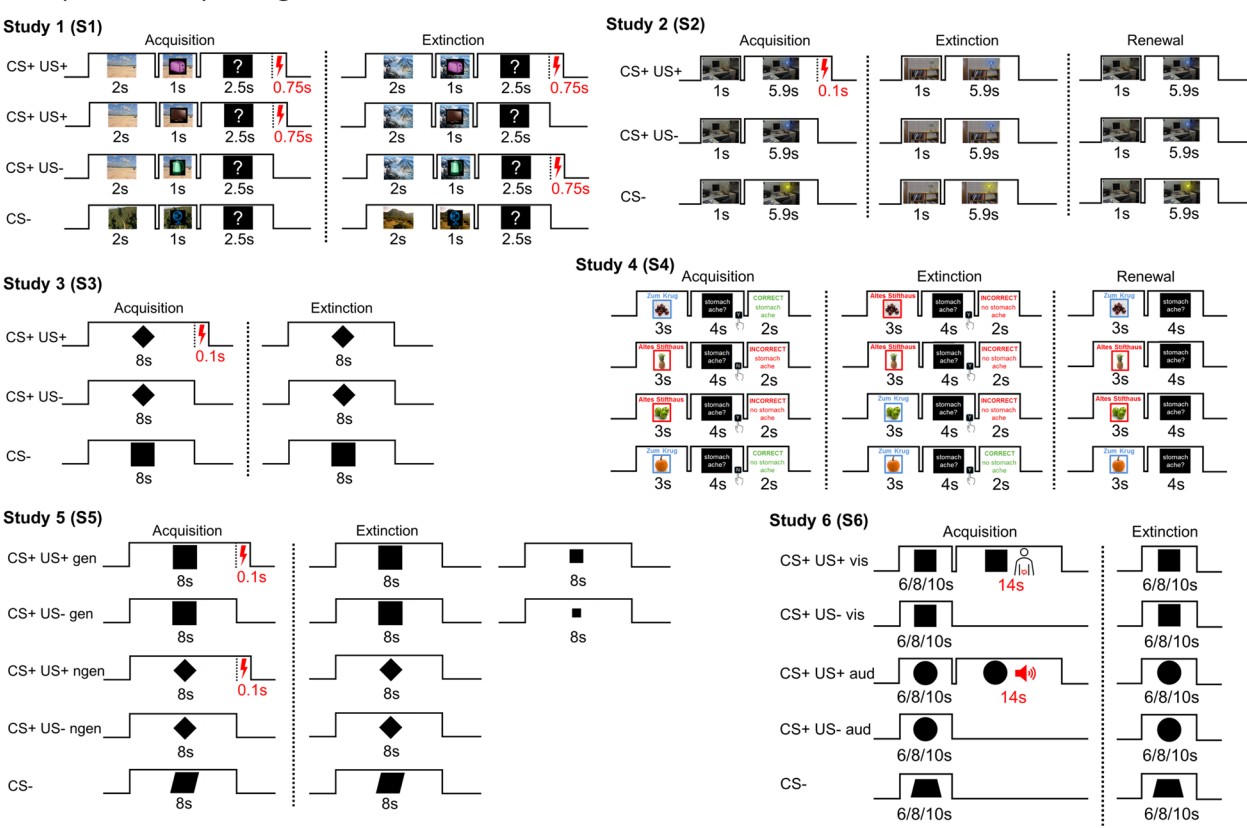

which some visual CS were partially reinforced by an aversive US (CS+; electrical shocks in S1–S3 and S5, visceral stimulation in S6), whereas other stimuli were never reinforced (CS−). The PL studies differed in that participants learned predictive associations between food items and a putative outcome ("stomach ache") across different contexts. All paradigms included acquisition and extinction phases, the latter consisting of unreinforced CS trials in a different context. Renewal was tested in S2 and S4 by showing the same stimuli in the acquisition context.

**Multimodal connectivity in the core learning network**

For the analysis of FC, we computed a composite score (concatenation of nine different metrics; see Methods, Supplemental Methods and Table S4) focusing on ipsilateral connections (e.g., left AMY – left HIP), with the exception of the CEB, given that (neo-)cerebellar regions are connected with the contralateral cerebral cortex. A mixed-effects model (participant nested within study) using FC as the outcome variable and ROI pair as predictor revealed greater FC for HIP-AMY than any other connection (all $zs > 18.56$, $ps_{FDR} < .001$, $ds > 1.18$, 95% CIs = [.35, 1.34]), followed by HIP-PFC (all $zs > 16.33$, $ps_{FDR} < .001$, $ds > .38$, 95% CIs = [.30, .79]) and AMY-PFC (all $zs > 4.67$, $ps_{FDR} < .001$, $ds > .30$, 95% CIs = [.03, 0.40]; see Table S9 for the remaining comparisons). This pattern was observed in both hemispheres (Fig. 2A).

The general pattern of SC was similar to what we observed for FC (Fig. 2B). A corresponding mixed-effects model using streamlines as the outcome variable revealed a disproportionate number of

**Fig. 1 | Number of datasets, regions of interest, and experimental paradigms. A** Number of resting-state fMRI (rs-fMRI) and diffusion-weighted imaging (DWI) datasets acquired in each group of the consortium [sfb1280.ruhr-uni-bochum.de]. Blue and red columns indicate whether learning was assessed via skin conductance responses (SCR) or behavioural ratings, respectively. Striped and plain columns reflect fear conditioning or cognitive predictive learning paradigms, respectively. **B** ROIs used in the present study. All subject-specific ROIs were extracted from an automatic parcellation and segmentation using FreeSurfer (top left). For the cerebellum, the three cerebellar nuclei (fastigial, interposed and dentate nuclei) were extracted using the SUIT package (Fig. S3; see Methods) and combined into one cerebellar ROI. For probabilistic tractography, surfaces of the dorsal anterior cingulate and ventromedial prefrontal cortices were used instead of their volumetric counterparts (Fig. S4). **C** Experimental paradigms of each group. S1: During acquisition, a context image was followed by either a CS+ item (partial reinforcement with US skin shock administered after 2.5 seconds, represented as a red lightning bolt) or a CS– item. During extinction, half of all items retained their contingencies, for the other half contingencies were reversed [Figure adapted from Bouyeure A. et al., 2025 (Study S1), published in eLife (https://doi.org/10.7554/eLife. 105126.3), licensed under CC BY 4.0 (https://creativecommons.org/licenses/by/4.

0/). Changes were made.]. S2: During acquisition, a context image was followed by either a CS+ item (partial reinforcement with US skin shock after 5.9 seconds) or a CS– item. During extinction, CS– and CS+ US- items were shown [Figure adapted from Milad M.R. et al. Biological Psychiatry 62, 446–454 (2007). Copyright © 2007, with permission from Elsevier]. S3: Similar to S2 but without a context image and with shapes instead of images as CSs. S4: Subjects indicated whether a food item predicted stomach ache during acquisition. During extinction, outcomes reversed for half of the food items [Figure adapted from Lissek S. et al. NeuroImage 81, 131–143 (2013). Copyright © 2013, with permission from Elsevier]. S5: Similar to S3 with the exception that CS+ US- items were presented in three different sizes during extinction. S6: During acquisition, distinct visual CSs were paired with either an aversive tone or a moderately painful rectal distension serving as US (represented by the red body symbol). During extinction, CS+ US- and CS– were shown. Figure 1C was created by the authors using Microsoft PowerPoint. *rs-fMRI* Resting-state functional magnetic resonance imaging, *DWI* Diffusion-weighted imaging, *SCR* Skin conductance response, *ACC* Dorsal anterior cingulate cortex, *AMY* Amygdala, *CEB* Cerebellar nuclei, *HIP* Hippocampus, *PFC* Ventromedial prefrontal cortex, *CS* Conditioned stimulus, *US* Unconditioned stimulus. Source data are provided as a Source Data file.

## A. Functional connectivity

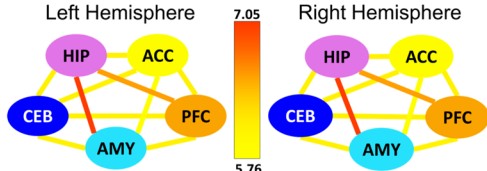

## B. Structural connectivity

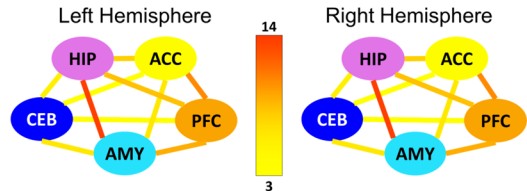

## C. Effective connectivity (average estimates)

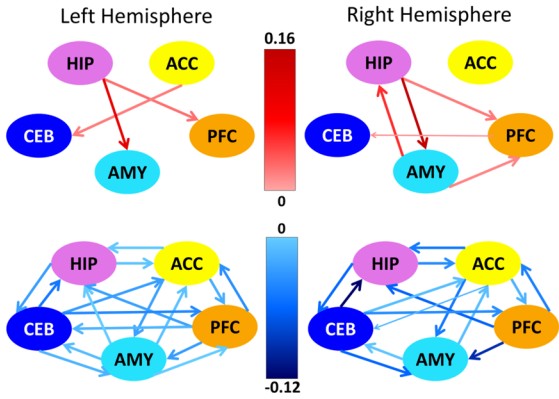

## D. Effective connectivity (Bayesian model averaging)

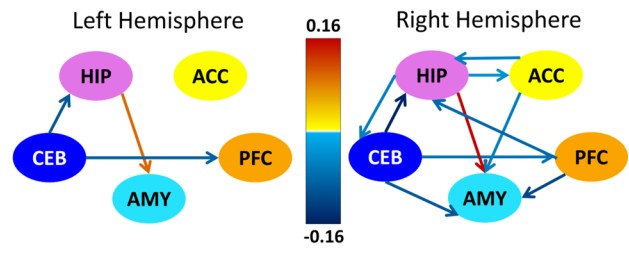

**Fig. 2 | Brain connectivity estimates. A** Average functional connectivity between all pairs of ROIs. Functional connectivity was calculated across the entire sample based on a composite metric (see Methods). Greater FC was observed for the connections HIP–AMY and HIP–PFC. **B** Average structural connectivity between all pairs of ROIs. SC values are based on streamline counts across the entire sample. As expected, the connection HIP–AMY showed a disproportionately larger number of streamlines, followed by ACC–PFC and AMY–PFC connections. **C** Average effective

connectivity based on spectral dynamic causal modelling (spDCM) estimates of directed connectivity among our ROIs (top: excitatory connections; bottom: inhibitory connections). **D** The winning model for effective connectivity showing the spDCM estimates of directed connectivity among our regions of interest computed using Parametric Empirical Bayes and Bayesian Model Averaging. *ACC* Dorsal anterior cingulate cortex, *AMY* Amygdala, *CEB* Cerebellar nuclei, *HIP* Hippocampus, *PFC* Ventromedial prefrontal cortex. Source data are provided as a Source Data file.

streamlines for the HIP–AMY, ACC–PFC, AMY–PFC, and HIP–PFC connections (in this order) relative to all others (all $z$s > 4.43, $p_{FDR}$ < .001, $d$s > .29, 95% CIs = [.09, 8.11]; see Table S10 for the remaining comparisons).

The EC analyses showed that the core learning network was mostly characterised by inhibitory connections, with only a few excitatory connections, most notably the bidirectional HIP–AMY connection (Fig. 2C; see also Fig. 2D for the group-level results using a Parametric Empirical Bayes model).

Since the same modelling approach was used for FC and SC, we could also compare the relative strength of connectivity for each connection between these two modalities (see Methods). This analysis showed that relative FC between HIP–PFC and AMY–PFC was greater than relative SC ($t(18957)$s > 8.30, $p_{FDR}$s < .001, $d$s > .06, 95% CIs = [.04, .25]), whereas for HIP–AMY, relative SC was greater than relative FC ($t(18957)$s > 17.15, $p_{FDR}$s < .001, $d$s > .12, 95% CIs = [.11, .29]). Thus, despite their overall similarities, relative FC and SC values differed for some ROI pairs (Fig. S11).

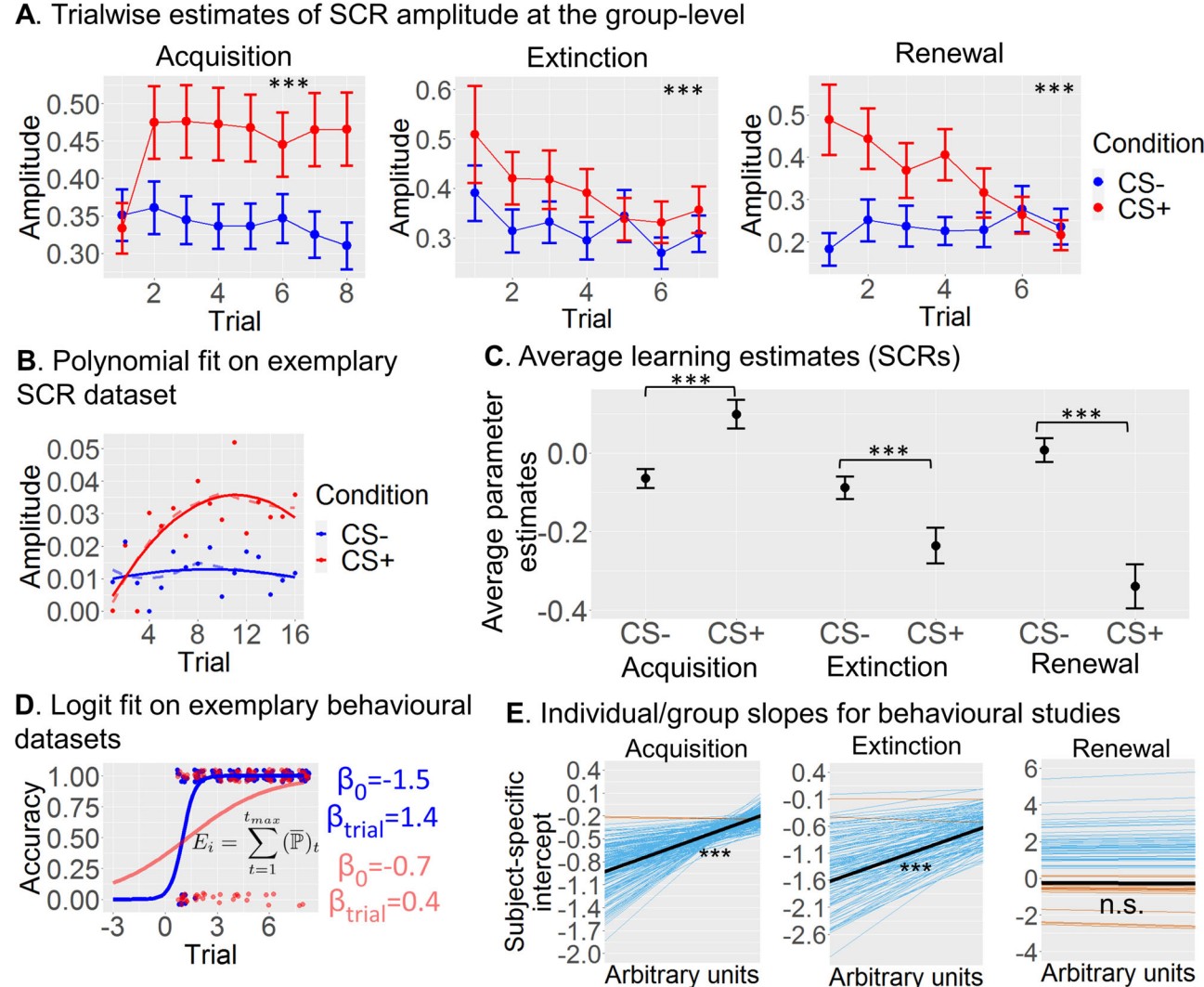

**Fig. 3 | Learning estimates. A** Group-level averages of SCR amplitudes in individual CS+ (red) and CS− (blue) trials during acquisition (left, n = 298), extinction (middle, n = 278) and renewal (right, n = 61). **B** Illustration of fixed-effect polynomial regression on the SCR data of an exemplar participant. After fitting the model, a unique learning score was computed comparing CS+ and CS− trials. **C** Average estimates of learning based on A and B for the entire sample (for n see (A); see Fig. S21 for the separate studies and Fig. S26 for different combinations of studies). **D** Multilevel generalised linear model using a logit link function to model behavioural ratings in predictive learning paradigms (exemplar participant). Individual parameter estimates were extracted from each participant and the expected rate of success after all 8 trials was computed. **E** Individual (blue: negative slopes; red: positive slopes) and group (black line) learning slopes estimated from the multilevel logistic regression in (D). *CS*+ Conditioned stimulus (reinforced), *CS*− Conditioned stimulus (non-reinforced), *SCR* Skin conductance responses. Data are presented as mean values +/- SEM. *** $p_{FDR}$ < .001, n.s. = not significant. All post hoc tests were two-sided and adjusted for multiple comparisons using FDR. Source data are provided as a Source Data file.

To examine whether the different types of connectivity were related to each other, we computed Pearson correlations between FC–EC, FC–SC, and EC–SC on each individual connection (Fig. S12). Interestingly, FC between several connections was significantly correlated with the respective EC patterns as well as with SC between these regions. By contrast, we did not observe any correlation between SC and EC connection strengths even at uncorrected thresholds. Thus, while individual differences in FC were partially determined by (putatively more hard-wired, i.e. trait-like) differences in streamlines, these SC differences did not correspond to individual differences in EC.

**Learning measures**
Learning during acquisition, extinction and renewal was estimated separately for each study and experiment. For studies that collected skin conductance response (SCR) data, PsPM was used to estimate trial-by-trial SCR values, while participant ratings were used in behavioural-only studies (see Methods).

For the group-level analysis, we included the initial eight trials (acquisition) and seven trials (extinction and renewal), which corresponded to the number of trials of the subject with the least number of trials. We observed a significant interaction of SCR data between time (i.e., trials) and condition (CS+ vs. CS−) for all experimental phases, indicating steeper increases in amplitude for CS+ vs. CS− during acquisition (t(4519) = 4.03, $p_{FDR}$ < .001, d = .34, 95% CIs = [.19, .49]) and steeper decreases for extinction (t(4010) = −3.26, $p_{FDR}$ = .001, d = .31, 95% CIs = [.12, .49]) and renewal (t(866) = −6.28, $p_{FDR}$ < .001, d = 1.02, 95% CIs = [.70, 1.34]) (see Fig. 3A). Initial CS+ responses during renewal were significantly elevated relative to late extinction and CS− trials, confirming the expected renewal effect (Fig. S39).

For the extraction of subject-specific variables of learning, extinction and renewal, we used all trials available in each participant in either a subject-wise polynomial regression for SCR data (studies S1, S2, S3, S5, S6; Fig. 3B) or a generalised linear mixed-effects model using a logit link function for behavioural ratings (study S4; Fig. 3D). Using

these individual-level estimates of learning, we observed a much larger estimate for CS+ than CS− in the acquisition phase, indicating greater learning for CS+ trials and the opposite pattern for extinction and renewal. Permutation testing on these learning scores confirmed that condition differences during all phases were significantly larger than would be expected by chance (Fig. 3C; acquisition: $t(292) = 3.34$, $p_{FDR} < .001$, $d = .20$, 95% CIs = [.06, .18]; extinction: $t(272) = −3.73$, $p_{FDR} < .001$, $d = .23$, 95% CIs = [.06, .18]; renewal: $t(60) = −4.99$, $p_{FDR} < .001$, $d = .64$, 95% CIs = [.21, .49]).

Similarly, in the behavioural studies, learning was highly significant at the group level during both acquisition ($z = 19.75$, $p_{FDR}s < .001$, $d = .37$, CIs = [.61, .75]) and extinction ($z = 16.11$, $p_{FDR}s < .001$, $d = .53$, CIs = [.84, 1.07]), with only 2 out of 180 individuals showing slight negative trends (see Fig. 3E). Not surprisingly, individual estimates of learning were again significantly above chance levels ($ts(177/185) > 49.92$, $p_{FDR}s < .001$, $ds > 3.66$, CIs = [6.47, ∞]; Fig. S15). Trial-by-trial changes in renewal were not significant ($z = −.74$, $p_{FDR} = .46$, $d = .06$, 95% CIs = [−.36, .17]), but the probability of making at least one renewal response (giving the same response as that given during acquisition) was still significant ($t(177) = 3.46$, $p_{FDR} < .001$, $d = .26$, 95% CIs = [1.23, ∞]). Further analyses showed that even though acquisition and extinction were correlated ($r = .16$, $p_{FDR} < .001$; Fig. S16), the amount of shared variance was very limited (≈ .03), suggesting that different factors may account for individual differences in these two phases. The correlation between acquisition/extinction and renewal was not significant, $rs = −.06/−.08$, $ps_{FDR} > .10$ (for correlations of learning estimates between FL and PL studies see Supplemental results and Fig. S16).

## Prediction of individual differences

We next investigated whether and how the three different types of connectivity (FC, SC, and EC) related to individual differences of learning using a LASSO regression model (see Methods).

### Acquisition

For acquisition, the relevant functional connections were lCEB–rPFC, lHIP–lPFC, rACC–rPFC, rAMY–rACC, rCEB–lHIP as well as bilateral HIP–ACC (Fig. 4A-B; using traditional p-values, these connections were all significantly above chance after correction for multiple comparisons, see Fig. S22). The results were similar for other combinations of individual studies (Fig. S27), and reliable over the course of several sessions (Fig. S30-S31) and varying scanning lengths (Fig. S33-S34). In addition to these functional connections that were relevant for learning across all studies, three connections were only predictive in PL studies (lACC–lPFC, lAMY–lPFC and lCEB–rHIP). Note that it is possible for a connection to be identified as significant when data are pooled across all studies, even if it was not significant in any individual study (see Fig. S23). Interestingly, when applied to canonical resting-state networks, our model showed that FC within the extinction network explained the most variance, with predictive performance increasing as spatial similarity to the extinction network increased (Fig. S29).

Neither structural nor effective connections significantly predicted learning for the entire sample. However, we did observe distinctive predictive connections for FL and PL paradigms when analysed separately. Regarding EC, disinhibition of the inhibitory connections lAMY→lPFC, rCEB→lAMY and bilateral HIP→AMY predicted acquisition in PL studies. In contrast, a more pronounced excitatory connection lCEB→rAMY predicted FL. For FLc, there was also an interesting association between acquisition and disinhibition of the inhibitory connection ACC→AMY, as well as increased PFC→AMY inhibition with greater acquisition. Thus, fear learning benefited from higher AMY inhibition by PFC and from AMY disinhibition by ACC. With regard to SC, only the connections lCEB–rPFC and rAMY–rPFC in PL were significant predictors.

Even though our results indicate that certain connections are predictive of acquisition performance, they do not reveal which regions play a predominant role. To assess the extent to which each of the five ROIs could act as a "hub" governing individual differences in acquisition, we reran the LASSO model on one hundred Monte-Carlo samples. In each iteration, we randomly selected 80% of the participants and recorded the frequency with which each ROI appeared in a significant connection. ROIs that appeared more frequently were considered more reliable and indicative of hub-like properties within the network. A Poisson mixed-linear model confirmed that, for functional connectivity (FC), all ROIs except the AMY were significantly different from zero ($zs > 2.73$, $ps_{FDR} < .05$, $IRRs > 1.17$, CIs = [1.04, 1.56]; AMY: $z = −1.47$, $p_{FDR} = .14$, $IRR = .92$, CIs = [.81, 1.03]). Post hoc tests revealed that ACC and HIP had larger numbers of appearances than the other ROIs ($zs > 2.25$, $ps_{FDR} < .05$, $IRRs > 1.13$, CIs = [1.02, 1.64]; see Fig. 4E, left panel). Regarding EC, only the AMY was significantly greater than 0 ($z = 5.75$, $p < .001$, $IRR = 1.26$, CIs = [1.16, 1.36]; other ROIs: $zs < 1.02$, $ps_{FDR} > .10$, $IRRs < 1.04$, CIs = [.83, 1.13]), showing a larger number of counts than the other ROIs ($zs > 4.33$, $ps_{FDR} < .001$, $IRRs > 1.21$, CIs = [1.11, 1.53]). Only the PFC was significant in the SC analysis ($z = 3.57$, $p_{FDR} < .001$, $IRR > 1.23$, CIs = [1.10, 1.38]).

In summary, the acquisition results indicated that FC was predictive of learning across the entire sample, with a strong focus on connections involving the ACC and HIP. Paradigm-specific EC- and SC-learning associations were also apparent, with differing dynamics for the PFC–AMY connections between PL and FL studies (disinhibition of AMY→PFC was beneficial for PL, whereas more pronounced inhibition of PFC→AMY predicted greater fear acquisition).

### Extinction

In stark contrast with our results during acquisition, we did not find any significant FC (or EC) connections that could predict individual differences in extinction learning across all paradigms. Interestingly, however, several structural connections consistently predicted extinction learning, most notably those involving the ACC, specifically, rHIP-rACC, lCEB-rACC, and bilateral AMY-ACC (Fig. 4A,C). Using traditional p-values, these connections were all significantly above chance after correction for multiple comparisons (Fig. S22). In addition to these effects across both FL and PL experiments, the structural connections lHIP–lPFC, rAMY–rACC and rHIP–rACC predicted extinction in FL but not PL studies. The results were similar for other combinations of individual studies (Fig. S28). However, the most interesting differences among the groupings were observed for EC, with a peculiar reversal in the direction of the AMY–HIP connectivity (benefit of more pronounced excitation of lHIP→lAMY for FL studies, benefit of higher disinhibition of lAMY→lHIP for PL studies), as well as a reversal in both directionality and valence of the HIP–ACC connection (less HIP inhibition by ACC being beneficial for PL extinction, and greater ACC inhibition by HIP being beneficial for FL extinction). In contrast, more pronounced PFC→ACC inhibitory connectivity was beneficial for extinction across the entire sample, as well as for FL separately.

Post hoc tests revealed that, for the SC analysis, the ACC had the largest number of appearances relative to all other ROIs ($zs > 8$, $ps_{FDR} < .001$; see Fig. 4E, middle panel). Regarding EC, the AMY, HIP, CEB and PFC were significantly greater than 0 ($zs > 2.34$, $ps_{FDR} < .05$, $IRRs > 1.10$, 95% CIs = [1.02, 1.19]; ACC: $z = 1.62$, $p_{FDR} = .11$), with the PFC showing a larger number of counts than any of the other ROIs ($zs > 3.66$, $ps_{FDR} < .001$, $IRRs > 1.15$, 95% CIs = [1.07, 1.23]). There was no significant ROI in the FC analysis.

In summary, extinction was mostly predicted by higher density of structural connections involving the ACC (and, to a lesser degree, AMY). Paradigm-specific learning was again predicted mostly by EC, specifically, a reversal in directionality and/or valence for the AMY–HIP and HIP–ACC connections.

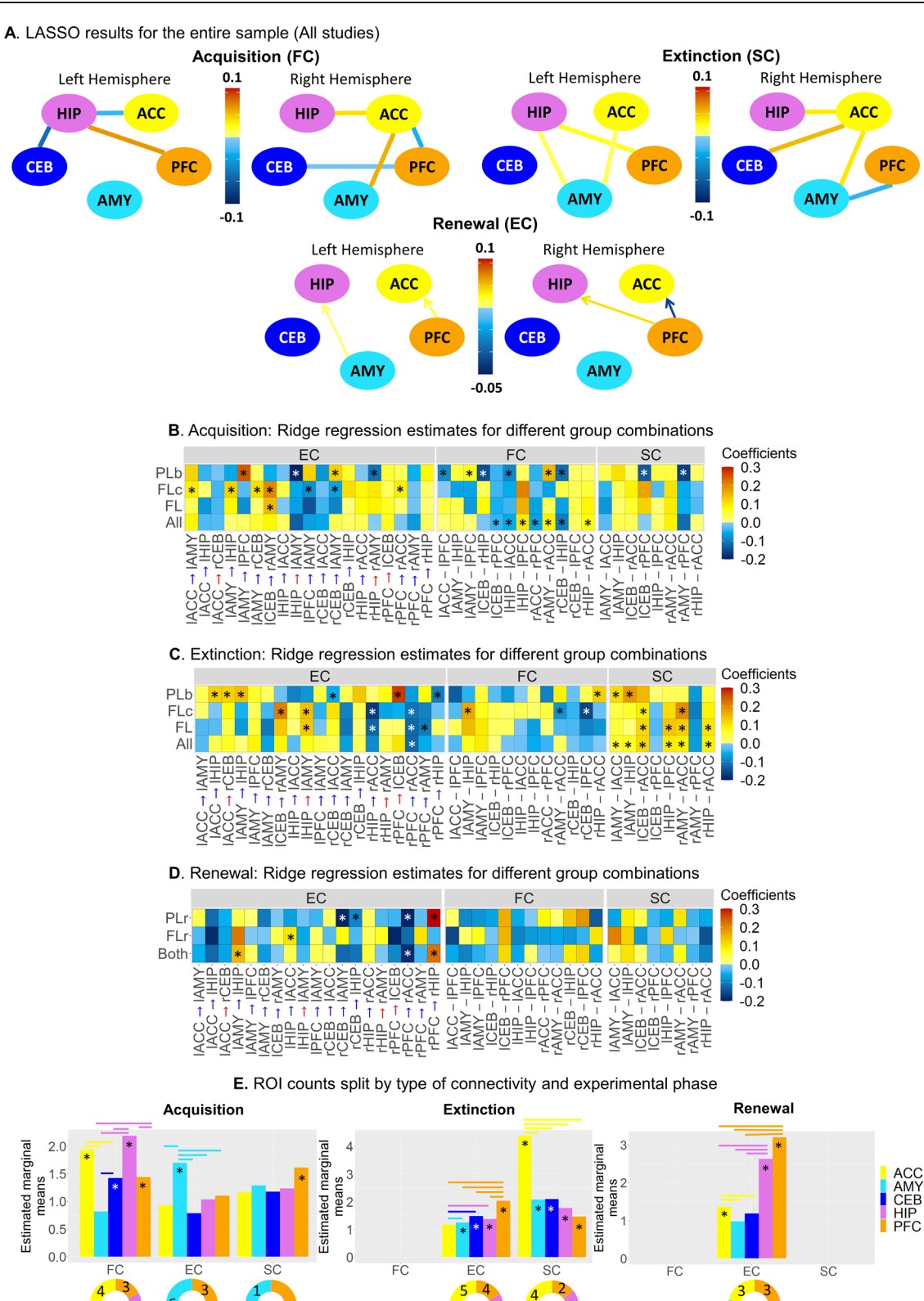

**A**. LASSO results for the entire sample (All studies)

**B**. Acquisition: Ridge regression estimates for different group combinations

**C**. Extinction: Ridge regression estimates for different group combinations

**D**. Renewal: Ridge regression estimates for different group combinations

**E**. ROI counts split by type of connectivity and experimental phase

## Renewal

Out of the six studies analysed above, S2 (FLr) and S4 (PLr) contained data about renewal after extinction. When both studies were analysed together, there were no significant FC or SC connections that could predict individual differences in renewal. However, the analysis of EC indicated that the greater the disinhibition of the HIP by AMY and PFC the greater the renewal effect, whereas renewal benefited from a higher inhibition of the ACC by the PFC (Fig. 4D; using traditional p-values, only rPFC→rHIP and rPFC→rACC were significantly above chance after correction for multiple comparisons, see Fig. S22). Separate analysis for FLr also revealed the relevance of disinhibition of the lHIP→lACC connectivity, whereas more pronounced inhibition of rCEB→lAMY, rCEB→lHIP, and rPFC→rACC connections, as well as disinhibition of rPFC→rHIP predicted renewal following PLr.

**Fig. 4 | Predicting individual differences from brain connectivity. A** Significant coefficient estimates from the LASSO model for the entire sample: FC only predicted acquisition, SC only predicted extinction learning, EC only predicted renewal. **B** Coefficient estimates were obtained using a ridge regression model for different groupings in the acquisition phase (see Fig. S23-S25 for lines of best fit for all connections). Ridge regression was chosen to provide estimates for all connections, as LASSO sets irrelevant coefficients to zero. Significant connections identified by LASSO are marked with asterisks (Ridge and nonzero LASSO coefficients were highly correlated, $r = .87$, $p < .001$). Only connections that were significant in at least one grouping for any connectivity type are displayed. **C** Same as (B) but for the extinction phase. **D** Same as (B) but for the renewal phase. In B-D, asterisks indicate which connections were nonzero in the LASSO model. **E** Bars represent the estimated marginal means for the number of times each ROI appeared in a significant connection across Monte-Carlo resampling iterations, where LASSO was run on random subsets of observations. This analysis assessed the robustness of each ROI as a "hub" within the fear and extinction network. Doughnut charts below each type of connectivity indicate how many times each ROI appeared in the original LASSO models (i.e., in B-D), providing a direct comparison between resampling-based estimates (bar plots) and the original results (doughnut charts). Asterisks indicate whether significant ROIs were greater than 0 ($ps_{FDR} = [.02 - <.001]$). Horizontal bars indicate significant differences between ROIs. All post hoc tests were one-sided and adjusted for multiple comparisons using FDR. *FC* Functional connectivity, *SC* Structural connectivity, *EC* Effective connectivity, *All* All studies (S1,S2,S3,S4,S5,S6), *FL* Fear Learning studies (S1,S2,S3,S5,S6), *FLc* Fear Learning classical paradigm (S2,S3), *PL* Predictive Learning studies (S4), *FLr* Fear Learning renewal study (S2), *PLr* Predictive learning renewal study (S4), *Both* Both renewal studies (S2, S4). *ACC* Dorsal anterior cingulate cortex, *AMY* Amygdala, *CEB* Cerebellar nuclei, *HIP* Hippocampus, *PFC* Ventro-medial prefrontal cortex. Source data are provided as a Source Data file.

The Poisson regression indicated that, for EC, only the ACC, HIP and PFC were significantly different from zero ($zs > 3.48$, $ps_{FDR} < .001$, $IRRs > 1.15$, 95% CIs = [1.06, 1.78]; other ROIs: $zs < 1.87$, $ps_{FDR} > .10$, $IRRs < 1.08$, 95% CIs = [.92, 1.16]). Pairwise comparisons revealed that the PFC had the largest number of appearances relative to all other ROIs ($zs > 2.61$, $ps_{FDR} < .05$, IRRs > 1.09, 95% CIs = [1.01, 1.85]), followed by the HIP ($zs > 7.91$, $ps_{FDR} < .001$, IRRs > 1.33, 95% CIs = [1.24, 1.70]; see Fig. 4E, right panel). No significant ROIs were observed for either the FC or SC analysis.

In summary, differences in EC were more sensitive than FC or SC in predicting renewal. Our results also indicated the importance of disinhibition of the HIP by both the PFC and AMY.

## Generalisability of learning predictors

We found that the models showed significant generalisability. When the LASSO model was trained on one type of paradigm (e.g., FL) and tested on the other type of paradigm (e.g., PL), we observed a higher generalisability (i.e., lower mean squared errors) relative to a surrogate model in which the learning estimates had been shuffled across participants while keeping the structure of the remaining data matrix intact (all $ps_{FDR} < .001$; Fig. 5A-C). Selecting non-overlapping groups for FL and PL led to identical results (Fig. S32).

In addition, we performed a leave-one-group-out (LOGO) cross-validation analysis (group here referring to the six individual studies), using a multiple regression analysis on the selected predictors. As Fig. 5D-F shows, the acquisition predictors were highly generalisable only when the functional connections were used (EC: $r = −.01$, $p_{FDR} = 1.00$; FC: $r = .14$, $p_{FDR} = .01$, SC: $r = −.01$, $p_{FDR} = 1.00$), whereas extinction predictors were only generalisable when the structural connections were used (EC: $r = .03$, $p_{FDR} = .23$; FC: $r = −.14$, $p_{FDR} = 1.00$, SC: $r = .23$, $p_{FDR} < .001$). For renewal, generalisability was found only when the effective connections were included, although this effect did not reach significance after correction for multiple comparisons (EC: $r = .12$, $p = .04$; FC: $r = −.02$, $p_{FDR} = .63$; SC: $r = −.02$, $p_{FDR} = 1.00$).

Follow-up analyses confirmed that for FC, prediction performance was indeed significantly greater for acquisition than for either extinction ($z = 3.77$, $p_{FDR} < .001$) or renewal ($z = 1.90$, $p = .03$), whereas SC prediction performance for extinction was significantly greater than for either acquisition or renewal ($zs > 3.2$, $ps_{FDR} < .01$). EC prediction performance showed a trend toward higher values during renewal compared to acquisition; however, renewal did not differ significantly from either acquisition or extinction after correction for multiple comparisons ($zs < 1.55$, $ps_{FDR} > .10$). We also compared prediction performance during acquisition, extinction, and renewal between the three types of connectivity. During acquisition, the FC prediction performance was greater than the performance of either SC or EC ($zs > 2.23$, $ps_{FDR} < .05$); during extinction, the SC prediction performance was greater than either FC or EC ($zs > 2.63$, $ps_{FDR} < .01$); and during renewal, the EC prediction performance showed a numerical difference from both FC and SC, although the effects did not reach significance after correction for multiple comparisons ($zs < 1.46$, $ps_{FDR} > .10$). Thus, our LOGO results are consistent with our results from the LASSO models in that, at the whole-sample level, acquisition, extinction learning, and renewal are best predicted by FC, SC, and EC, respectively.

To the extent that the functional architecture of the individual's brain is intrinsic, we should be able to observe a correspondence in the FC profile between task-based and resting-state connectivity[27]. Given that, in our study, resting-state FC in seven distinct connections were shown to predict acquisition across the entire group (Fig. 4A-B), we queried whether task-based FC would show a similar association. For that purpose, we selected two experiments from S2 for which task-fMRI data were available (N = 137). Task-based FC was computed for the time periods corresponding to the expected peaks of the BOLD signal for both CS+ and CS− trials during the acquisition phase. The difference in FC between CS+ and CS− was then calculated for each of the relevant ROIs and used as predictors in a subsequent multiple regression model (see Fig. 5G and Methods).

The model including the LASSO predictors was significant ($F(7,90) = 2.39$, $p = .028$, $R^2 = .16$; Fig. 5H). In addition, three out of six predictors reached individual significance (rHIP–rACC: $b = 0.18$, $p = .024$, 95% CI = [0.02, 0.33]; lHIP–lACC and rCEB–lHIP: $bs < −0.14$, $ps < .05$, 95% CIs = [−0.31, −0.02]; see Fig. 5I). This result indicates that task-dependent functional connectivity during acquisition, using the predictors from the resting-state LASSO model, was predictive of learning performance as well. This result was specific, as none of the models using pseudo-random functional connections within the core learning network were significant (all $Fs(7,90) < .32$, $ps > .05$, $R^2s < .09$), and the maximum $R^2$ value that could be achieved (0.089) was almost half of that for the significant model with the LASSO predictors (0.157; see Fig. 5H).

Next, we enquired whether our combination of selected predictors would show a performance advantage over random connections between ROIs that were not selected in the original LASSO models, by using simulated data. If so, this would show an additional layer of generalisability in the sense that our predictions could potentially be applicable to new datasets. For that purpose, we artificially generated 100 independent datasets (including all predictors, learning measure and covariates), each with 40 observations, based on purely simulated data drawn from a multivariate normal distribution that mimics the relationships between variables in our original dataset (see Methods). The performance of the selected LASSO model was tested against a pseudo-random model that consisted of an equal number of predictors to the LASSO model but chosen at random (excluding the LASSO predictors) within the same type of connectivity.

Figure 5J-L show the results of the simulation analysis (see also Fig. S35 for the results using bootstrapping). For both acquisition, extinction, and renewal, we observed significantly larger $R^2$ values for

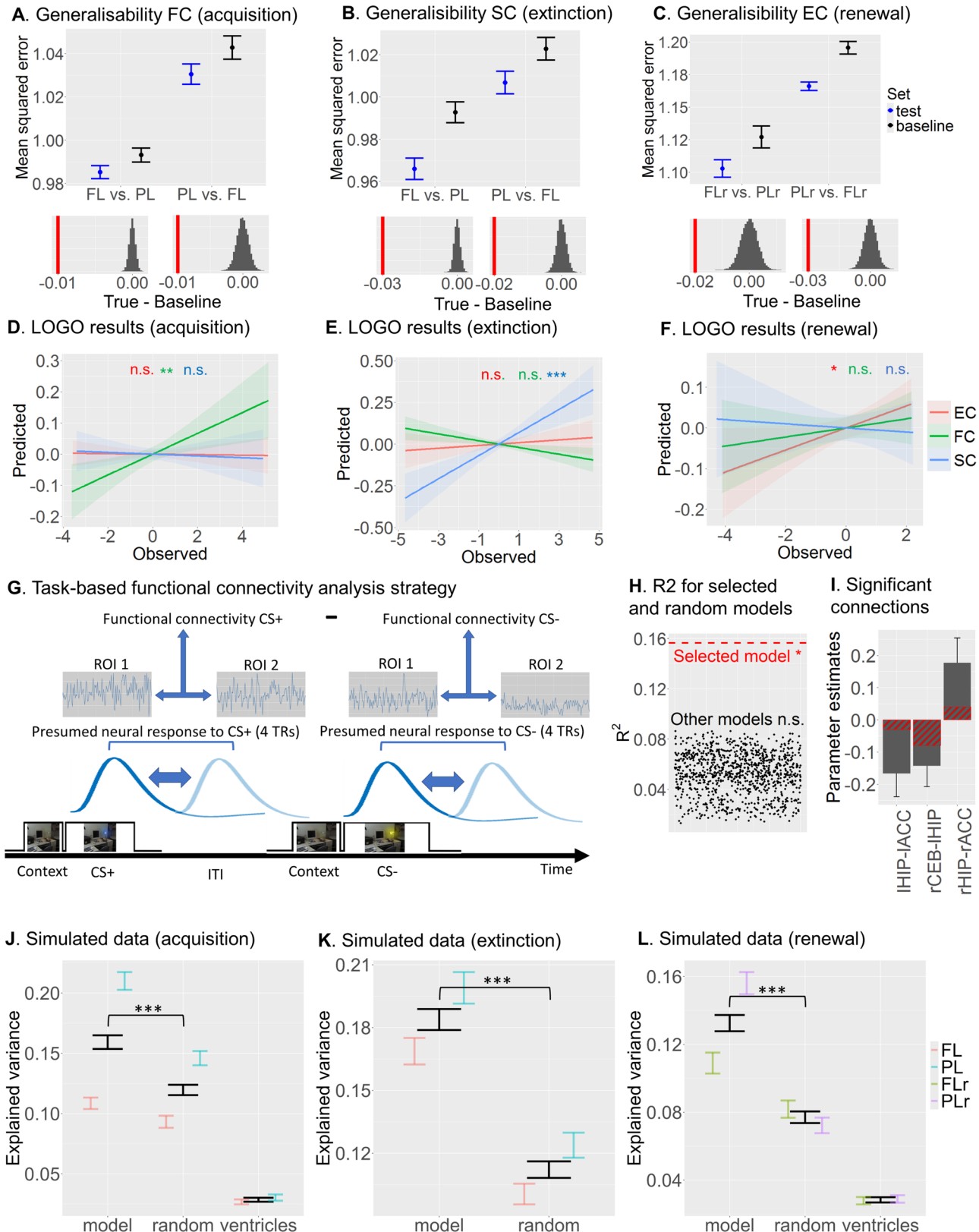

**A.** Generalisability FC (acquisition)

**B.** Generalisibility SC (extinction)

**C.** Generalisibility EC (renewal)

**D.** LOGO results (acquisition)

**E.** LOGO results (extinction)

**F.** LOGO results (renewal)

**G.** Task-based functional connectivity analysis strategy

**H.** R2 for selected and random models

**I.** Significant connections

**J.** Simulated data (acquisition)

**K.** Simulated data (extinction)

**L.** Simulated data (renewal)

the LASSO model relative to the pseudo-random model (all $ts(198) > 6.81$, $ps_{FDR} < .001$, $ds > .96$, 95% CIs = [.03, .08]; black bars in Fig. 5J-L). In addition, when the data were split into an FL and a PL group, better performance of the LASSO models was again observed for both groups for acquisition, extinction and renewal (PL: $ts(198) > 6.80$, $ps_{FDR} < .001$, $ds > .96$, 95% CIs = [.05, .10]; FL: $ts(198) > 2.22$, $ps_{FDR} < .05$, $ds > .31$, 95% CIs = [.002, .08]; coloured error bars in Fig. 5J-L).

In summary, generalisation of our predictions occurred between FL and PL paradigms, but only for the types of learning that could be predicted by the particular type of brain connectivity (i.e., transferable acquisition, extinction and renewal effects only for FC, SC, and EC, respectively). In addition, our (resting-state-based) model connections were more successful in predicting learning from task-based FC than alternative connections. Finally, our model predictions were also more

**Fig. 5 | Generalisability of learning and simulations. A-C** Generalisability of acquisition (**A**, n = 401), extinction (**B**, n = 349), and renewal (**C**, n = 201) was examined by training a LASSO model on one type of paradigm (e.g., FL) and testing it on the other type (e.g., PL). Statistical significance was assessed using two-sided permutation tests with FDR correction for multiple comparisons (see histograms; red vertical lines indicate the difference between the true and the baseline means). Mean squared errors (MSEs) of each of the models were used as measure of fit. **D-F** Leave-one-group out (LOGO) cross-validation indicated that functional, structural and effective connectivity predictions were generalisable for acquisition **D**, extinction learning **E** and renewal **F**, respectively. Prediction performance was quantified as the Pearson correlation between observed and cross-validated predicted values. **G** Strategy for computing task-based functional connectivity for study S2. *CS+* Conditioned stimulus (reinforced), *CS−* Conditioned stimulus (non-reinforced), *ITI* Inter-trial interval, *TR* Repetition time. **H** A multiple linear regression analysis showed that significant functional connectivity was obtained for our LASSO model but not for other models using random connections (p-values

uncorrected for multiple comparisons). **I** Significant connections obtained from the significant model in **F**, n = 90. Red stripes indicate the estimates for those connections in the resting-state LASSO model. **J-L** Simulation analysis indicated that our model predictions were superior in predicting acquisition (**J**, n = 401), extinction learning (**K**, n = 349) and renewal (**J**, n = 201) relative to either the same number of random connections from the model (random), or connectivity between the 4th ventricle and lateral ventricles (ventricles). In A-F and I-L, data are presented as mean values +/- SEM. *FC* Functional connectivity, *SC* Structural connectivity, *EC* Effective connectivity. *FL* Fear Learning studies (S1,S2,S3,S5,S6), *PL* Predictive Learning studies (S4), *FLr* Fear Learning renewal study (S2), *PLr* Predictive learning renewal study (S4). *ACC* Dorsal anterior cingulate cortex, *CEB* Cerebellar nuclei, *HIP* Hippocampus. * $p$ = .04 (uncorrected), ** $p_{FDR}$ = .01, *** $p_{FDR}$ < .001, n.s. Not significant. All post hoc tests were two-sided and adjusted for multiple comparisons using FDR (unless specified otherwise). Source data are provided as a Source Data file.

successful in predicting learning in simulated datasets than pseudo-random connections within the core learning network.

## Discussion

Here we show that individual abilities of learning, extinction, and renewal can be explained by distinct types of resting-state connectivity among a small set of regions within the core learning network. This is substantiated by the observation of a triple dissociation, with FC, SC, and EC better predicting acquisition, extinction learning, and renewal, respectively, highlighting the distinct functional roles of different types of brain connectivity across learning phases. Our findings reveal a core role of the ACC as a "hub" within this network. Surrounding this core, additional areas make specific contributions to learning, extinction, and renewal.

Contrary to previous assumptions, we demonstrate that heterogeneity in learning measures, experimental setups, MRI sequences, and statistical methods are not a detriment for reliably detecting across-study effects, provided that careful harmonisation of sequence parameters and appropriate statistical modelling are employed. Indeed, the results of our generalisation analyses make it likely that our findings are applicable to a great variety of learning paradigms. Thus, we believe that the present study represents a major step forward in identifying a global neural architecture that determines individual abilities of learning and extinction, as well as the propensity for renewal. We will outline this in the following sections, one by one.

## Different types of connectivity within the core learning network

Our results, shown in Fig. 2A-D, demonstrate that the three types of connectivity – expressed by structural, functional and effective connectivity – are clearly dissociable, suggesting that they reflect different neurophysiological processes. For example, HIP–AMY connectivity was noticeably higher than any other connection, which is not surprising given the myriad animal fear learning studies showing a tight coupling between these two regions (for reviews see[12,14,15]). Importantly, however, the relative connectivity strengths were not always equivalent between FC, SC, and EC – for instance, ACC and PFC showed stronger structural connections than HIP-PFC, even though FC was higher for the latter connection. Similarly, cerebellar effective connectivity to HIP, AMY, and PFC showed some of the strongest effects, even though they shared the least number of streamlines (see Fig. 2B). This dissociation between the three types of connectivity is the basis for their putatively different functional implications.

The EC results revealed mostly inhibitory connections, with the notable exception of the HIP→AMY excitatory connection. Given that long-range connections are assumed to be predominantly excitatory[42], our finding that most extrinsic (inter-areal) connections were inhibitory may appear puzzling. However, recent evidence suggests that a

homeostatic brain makes abundant use of inhibitory connectivity, which allows for an effective control of the stability of memory patterns[43,44]. One mechanism contributing to this inhibitory network is through feedback-driven regulation.

### Neural predictors of acquisition

Fear acquisition involves a conditioned stimulus (CS) that does not elicit any response on its own and an unconditioned stimulus (US) which elicits strong responding without the need for prior training. Our analyses of FC patterns show that pre-existing functional inter-actions reliably predicted the amount of learning across the highly diverse fear conditioning and predictive learning paradigms that we employed. Since FC is strongly state-dependent[45], our results point towards the relevance of pronounced intra-individual differences in learning. This is substantiated by the observation that acquisition could not be predicted by arguably more stable and trait-like patterns of structural connectivity. The gradually increasing learning curve observed across participants often results in the misleading interpretation that an association is learned slowly and at similar speed. However, individual learning curves often reveal a step-like rise of underlying associative strength that occurs with high differences[46]. Our results suggest that intra-individual differences captured by FC contribute to this variance.

Fear conditioning and predictive learning are known to go along with ACC activation[23,24]. Accordingly, we found that ACC connectivity to HIP, PFC, and AMY were among the most reliable predictors of acquisition across the entire sample (see Figs. 4B and 6A). The HIP has been commonly linked to the formation of new episodic memories[47], whereas connectivity with the PFC may reflect recruitment of appraisal processes[16]. Furthermore, projections from ACC to AMY appear essential in mediating fear behaviour[48], and our results suggest that this effect also applies to PL. Thus, the functional relevance of HIP–ACC, ACC–PFC, and AMY–ACC connectivity may be related to the acquisition of novel memories and the evaluation of the motivational significance of the CS.

The connection to CEB might have a different character. During fear learning, large parts of the cerebellar cortex are activated by the prediction error[17]. Firing patterns of the cerebellar fastigial nucleus regulate fear-learning via thalamo-prefrontal dynamics, freezing behaviour through the periaqueductal grey, and anxiety behaviour via thalamo-amygdala systems[49]. Given that during Pavlovian conditioning, the CS usually precedes the US with high contiguity, the CEB could serve as a fine-tuned predictor for the timepoint of the occurrence or the absence of the US.

In summary, fear-related information from cortico-thalamic pathways may arrive in the AMY for initial processing ("quick and dirty" route[50]). AMY connection to ACC and onwards to PFC could have important roles in appraisal and, for FL, suppress excess AMY activity,

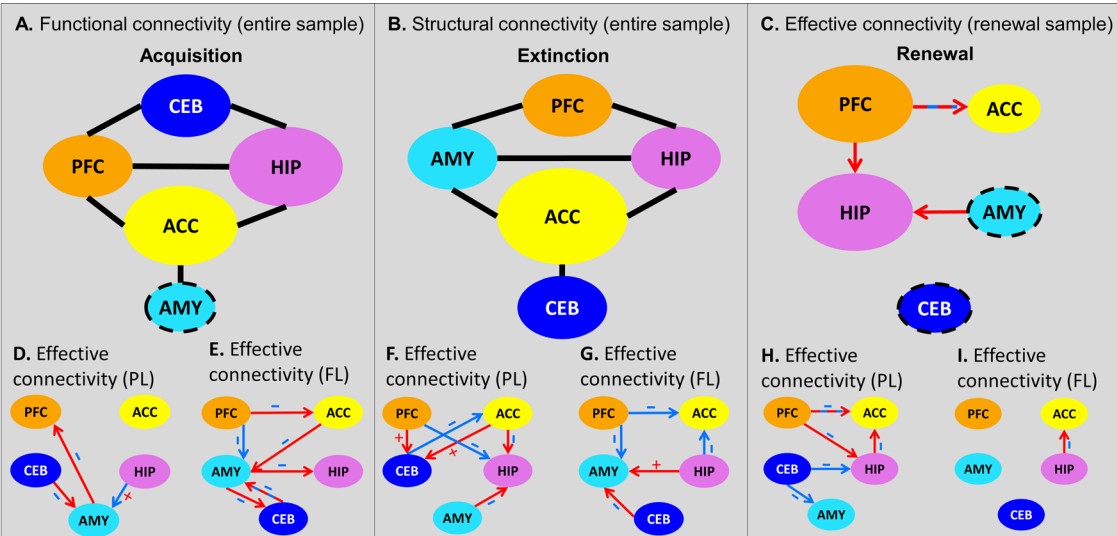

**Fig. 6 | Graphical depiction of all models in which significant connections between nodes were found. A** Summary of the acquisition model using the combined sample, showing the significant functional connectivity connections. The presence of a connection reflects its selection by LASSO in Fig. 4. For example, in Fig. 4B, LASSO identified a relevant ACC–HIP FC connection but not an ACC–CEB FC connection, which is depicted here as a line connecting ACC–HIP but not ACC–CEB. The size of each node represents its relative importance, as suggested in Fig. 4E, with dashed outlines indicating non-significant nodes. **B** Same as (A) but showing the structural connectivity connections significant in the extinction model. **C** Same as (A) but showing the effective connectivity connections significant in the renewal model. **D,E** Graphical depiction of the PL (D) and FL (E) model showing the effective connectivity connections significant in the acquisition model. **F,G** Same as (D,E) but for the extinction model. **(H-I)** Same as (D,E) but for the renewal model. The panels D-I integrate results from spectral DCM and LASSO. Red and blue arrows indicate whether the association between learning and connectivity was positive or negative, respectively (see Fig. 4B-D). Connections with alternating red and blue arrows indicate that their effects varied by hemisphere. Plus and minus signs indicate excitatory and inhibitory connections, respectively (see Fig. 2C). Red arrows on inhibitory connections (- signs) indicate that disinhibition of these connections was beneficial in a given learning phase, while blue arrows on inhibitory connections (- signs) indicate a benefit of more pronounced inhibition. For example, in Fig. 6E, the AMY→HIP connection is shown as a red arrow with a minus sign. According to Fig. 2C, this connection is inhibitory (blue), which is indicated by a - sign in the current figure, while the LASSO model for acquisition found a positive coefficient for this connection (red arrow). Thus, this figure visually conveys which connections are inhibitory (- signs) and whether learning benefited from reduced inhibition (red arrows) or increased inhibition (blue arrows). *ACC* Dorsal anterior cingulate cortex, *AMY* Amygdala, *CEB* Cerebellar nuclei, *HIP* Hippocampus, *PFC* Ventro-medial prefrontal cortex, *PL* Cognitive predictive learning, *FL* Fear learning.

while HIP and CEB are implicated in the formation of new memory traces and indicating time points of prediction error information, respectively. These types of information may be relayed to the ACC, which, in turn, conveys this information back to the AMY for final integration and gating of fear responses.

### Neural predictors of extinction learning

During extinction, the CS is no longer followed by the US. This ignites an expectancy-driven prediction error that does not erase the CS–US acquisition memory but establishes a second inhibitory trace that suppresses the occurrence of the conditioned response[2]. These learning events were only predicted by SC across the entire sample, but not by FC or EC. This suggests that the propensity of extinction is contingent on stable individual traits. Consequently, extinction learning is related to personality traits, including tolerance of uncertainty[51] and trait anxiety[52], and is particularly sensitive to rather stable microstructural white matter measures and cortical thickness of selected emotional circuits[52,53].

As with acquisition, the ACC played a central role in extinction learning. Not only did its connectivity with HIP, AMY and CEB predict the speed of extinction, but also the connections involving the ACC were by far the most reliable. Indeed, two recent human neuroimaging meta-analyses reported activation of this region as the most consistent finding during extinction[25,54], suggesting the ACC may function as a "hub" within the core learning network[55]. In addition to ACC, an intact HIP–PFC pathway seems crucial for the formation (and recall) of fear extinction and its contextual modulation in animal[15] as well as human neuroimaging studies[56]. Furthermore, PFC–AMY connectivity predicted extinction learning, corroborating early human neuroimaging

connectivity findings[18,19] as well as results from rodents and non-human primates[57,58]. The integrity of the PFC–AMY pathway may thus be critical in allowing the PFC to regulate the formation and maintenance of extinction memories by active suppression of AMY output[14]. Finally, extinction could be predicted by HIP-AMY SC. Given their roles in the encoding of contextual representations and expression of CRs, respectively[14,15], a stable HIP–AMY pathway would ensure that their integration results in a contextually-appropriate response[59]. Our results could imply that the extent of this ability is a trait factor.

Regarding EC, analysis of FL indicated that inhibition of ACC by both PFC and HIP was related to extinction. Also, just as with acquisition, PFC suppression of AMY activity improved fear extinction in FL studies. We also found a modulation of AMY activity by HIP input (see Fig. 6G). Since extinction learning is context-dependent, the excitatory HIP→AMY pathway would enable the HIP to relay back the crucial CS–context information necessary for the formation of context-dependent extinction memories in the AMY.

In sum, our results indicate that structural connections among PFC–HIP–AMY may form part of a circuit in which the PFC suppresses excess AMY excitation (for FL only) under the HIP-driven representation of the appropriate context (for both FL and PL). The ACC may be a critical "hub" that synchronises these interactions and delivers the output to AMY for final integration.

### Neural predictors of renewal

As outlined above, extinction is not an erasure of the old association, but rather the formation of a new memory trace of an inhibitory nature. This can be demonstrated by several phenomena of which renewal is possibly the most interesting one. Renewal is the recovery of the

extinguished response, induced by changing the context from that of the extinction phase back to that of acquisition. By this, it becomes obvious, how much context-dependent extinction learning is[2,60]. Across all studies, individual differences in renewal could only be explained by EC. Even though we cannot discard the possibility that absent SC (or FC) associations may partially relate to lower statistical power (only studies S2 and S4 were included), the EC results nevertheless indicate that the inhibitory network (see Fig. 6H-I) conveys more information about renewal than that afforded by either SC or FC. Also note that rs-fMRI was recorded shortly before acquisition/extinction, whereas renewal was tested the day after. If renewal is related to state variability, and if FC is more susceptible to decay over time than EC[61], this could explain the prevalence of EC-related associations with renewal.

The communication between HIP and PFC was particularly involved in renewal. Indeed, fear renewal increases connectivity between HIP and PFC[35], inactivation of either of these two components reduces fear renewal[62,63], and both PFC and HIP activity during extinction is positively correlated with renewal in predictive learning[21–23]. Renewal is also strongly context-dependent. A recent computational study made it likely that hippocampal replay is necessary and sufficient to generate context representations in the PFC[64]. Indeed, only participants with stronger hippocampal activation during extinction in a novel context relative to the acquisition context show renewal[65]. Accordingly, our PL studies showed that greater renewal was associated with HIP disinhibition by the PFC.

Also, the unidirectional AMY→HIP connection was relevant for renewal. In rodents, AMY projections to ventral HIP modulate affective states[66,67], such that this pathway may play a major role in the reemergence of fear memories[59]. Indeed, renewal probably represents a re-activation of the neuronal ensemble that was constituted during acquisition[60]. Simultaneous recordings from amygdala and hippocampal CA1 in fear-conditioned mice show synchronised activity that is related to the fear associated CS and results in freezing[68]. Thus, synchronised AMY→HIP connections could mediate fear memory retrieval that is associated with the re-exposure of the contextual cues that were present during acquisition but absent during extinction. This fits with the observation that a specific activation of a sparse ensemble of hippocampal cells elicits the recall of a context-bound memory engram of fear[69].

In summary, our results indicate that renewal is driven by the interaction of PFC and HIP, possibly by hippocampal replay that generates representations of the critical context in the PFC. Additionally, AMY input to HIP possibly mediates the activation of fear-related memories in context-dependent ways. This could imply that prefrontal fear memories become contextually bound by hippocampal replay and can subsequently be activated during renewal by AMY activation that processes the context that was present during acquisition.

## Generalisability and clinical relevance of our findings

Across three different types of generalisability analysis, we were able to show that our model predictions are not only transferable between FL and PL studies, but also that the connections identified during resting states predict learning even when they are applied to task-based FC. A triple dissociation using across-studies predictions confirmed the higher predictive power of FC for acquisition, SC for extinction, and EC for renewal. The simulation results also indicate that our model could potentially be applied to new data – using our selected connections resulted in more explanatory variance than using pseudo-random connections from the same model.

The generalisability of our findings may also have clinical implications for psychotherapy practice. Anxiety disorders are typically treated within a particular therapeutic context, so one major challenge is to ascertain how to reduce fear over and above the therapeutic setting. To translate basic research to the treatment of fear-related

disorders, it is crucial to understand how fears are both acquired and inhibited. We believe our study is a step closer toward that aim. Across a large group of (healthy) participants and different paradigms, we showed that fear and extinction learning can be reliably predicted by only a few connections between selected brain regions. The finding that the propensity for extinction may be more hardwired than renewal could imply that rather than optimising a person's extinction ability, it may be more efficient to target (i.e., prevent) renewal.

Fear learning and extinction are putative core mechanisms in the psychopathology of, and treatment for, affective disorders. Thus, our findings may provide a foundation for future research into how individual differences in connectivity patterns related to learning and extinction contribute to differences in the risk for (or resilience against) affective disorders, and inform potential therapeutic interventions. Thus, our results complement current research in precision medicine on the role of different resting-state networks for distinct biotypes of affective disorders (including depression), which may help guide decisions about specific therapeutic interventions[29]. Specifically, they indicate that structural connectivity is more directly related to extinction learning – and thus possibly to exposure therapy – than functional connectivity, and suggest that the individual strengths of structural network connections should be considered in clinical populations in addition to currently applied measures of functional connectivity[70].

Finally, our data show that both acquisition and extinction involve extensive interactions with PFC and ACC. Indeed, research on the neural bases of cognitive-behavioural therapy (CBT; the most popular psychotherapy treatment for anxiety-related disorders) has suggested that therapy success depends on stable inhibitory control of the AMY by the PFC and ACC[71]. Thus, modulating the PFC/ACC during extinction (e.g., by altering schema representations in these regions) could potentially affect subsequent novel fear learning. This could be achieved by using non-invasive techniques such as repetitive transcranial magnetic stimulation (rTMS) and/or theta band transcranial alternating current stimulation (tACS), both of which have been shown to specifically modulate PFC/ACC function[55,72,73].

## Limitations of the present study

One important limitation of the present study was the relatively small number of ROIs in our network. The inclusion of only a subset of regions was driven not only by theoretical reasons (see Supplemental Fig. S36 and Table S1), but also a statistical one. When computing EC, the number of possible ipsilateral connections is given by the formula $2 \times (n^2 - n)$. Thus, the addition of only two ROIs would have more than doubled the number of predictors in our models, surpassing the number of participants in some of the experiments. Future studies could combine our brain connectivity findings while exploring additional connections which, together, may enhance predictive power.

All experiments analysed here fell under either the FL or the PL paradigm. However, FL studies consisted of relatively heterogeneous experiments (e.g., with different stimuli, trial numbers, and US types), whereas PL studies were more homogenous. On the other hand, the heterogeneity of FL studies also allowed us to find common predictors that generalised across a broad range of experimental conditions, which would not have been possible when only a single paradigm had been conducted. In this vein, future work may aim to include an even greater variety of paradigms, such as eyeblink conditioning, appetitive, and olfactory learning.

In addition, the lack of symmetry in explaining learning from the same connection in both hemispheres was surprising, given that the connectivity estimates were similar between hemispheres. This result may have partly been due to an inherent feature in LASSO's variable selection procedure. Left- and right-hemispheric connections are often highly correlated; LASSO may thus favour the connection from one hemisphere which is a slightly better predictor than the

corresponding connection from the other hemisphere, and penalise the other, leading to asymmetries in the identified contributing connections. Even though this feature does not directly affect the conclusions reached in our study, future work could explore the impact of different regularisation strategies to better account for the relative contributions of both hemispheres in predictive modelling.

Finally, it should be mentioned that the generalisability results showed numerically small (albeit significant) improvements over baseline (e.g., differences in the order of .03 in Fig. 5A-C), although it was slightly higher for the LOGO, task-fMRI, and simulation analyses, where estimates ranged between .07–.23. We cannot discard the possibility that this modest transference between paradigms may be the consequence of using a somewhat restricted network as well as low explained variance overall. Indeed, rs-fMRI usually explains only modest variance in behavioural data even with large numbers of ROIs (2%–10%[28,74]), which likely places a ceiling on the maximum achievable generalisability across paradigms. Future studies involving different paradigms may wish to control for task demands and temporal/contextual characteristics of learning, as well as employing a broader network to increase the chances of generalisability

Individual differences in learning and extinction are core determinants of cognitive flexibility and are believed to be crucial for explaining treatment success of fear-related disorders. Despite its obvious fundamental relevance and clinical importance, the lack of consensus regarding the neural mechanisms of underlying individual abilities in learning and extinction has hindered progress in the translation of neurobiological models of extinction into clinical applications. We believe our study takes a step forward in bridging that gap. Through a careful process of homogenisation, using approaches to identify subject-specific learning across different paradigms, applying complementary types of brain connectivity, and conducting state-of-the-art statistical modelling, we were able to show both shared and distinct neural mechanisms of learning and extinction across a multitude of paradigms. These results have profound implications for understanding why the abilities of learning and extinction, as well as the propensity to show renewal, are highly variable in both healthy and clinical populations.

## Methods

*Participants.* The study described here was conducted as part of a large-scale collaborative research project, SFB1280 "Extinction Learning" (sfb1280.ruhr-uni-bochum.de). Participants were recruited for different studies within this project. In addition to task-based fMRI, participants in these studies also took part in resting-state and/or diffusion-weighted imaging scanning sessions. Only participants with neuroimaging data from at least one of these modalities were included in the present study (see Fig. 1A). The individual studies contributing to this project were approved by the Ethics Committee of the Medical Faculty of the University of Duisburg-Essen (Ethik-Kommission der Medizinischen Fakultät der Universität Duisburg-Essen) and by the Ethics Committee of the Ruhr University Bochum. All participants gave written informed consent and were monetarily reimbursed or received course credits (see Table S2 for demographic information).

For resting-state fMRI, 509 participants in total were included in 6 different studies: S1, N = 28 [age = 24.4 (3.51 SD), 19 women]; S2, N = 151 [age = 22.0 (2.20 SD), 95 women]; S3, N = 44 [age = 23.5 (3.56 SD), 22 women]; S4, N = 176 [age = 25.8 (4.19 SD), 89 women]; S5, N = 55 [age = 24.1 (3.74 SD), 26 women]; S6, N = 55 [age = 26.3 (4.66 SD), 38 women].

For diffusion-weighted imaging, 467 participants in total were included in 5 studies: S2, N = 165 took part in S2 [age = 21.9 (2.17 SD), 103 women]; S3, N = 44 [age = 23.5 (3.56 SD), 22 women]; S4, N = 174 [age = 25.7 (4.04 SD), 85 women]; S5, N = 25 [age = 24.1 (3.55 SD), 11 women]; S6; 55 [age = 26.5 (4.85 SD), 27 women].

### Scanning sequences

All MRI images were acquired using a 3T MRI scanner (S1, S2, S4, S5, Philips Achieva; S3, Siemens MAGNETOM Vida; S6, Siemens Skyra). Participants were scanned at three different locations using three different 3T MRI systems from two different vendors, so distinct sequence and imaging parameters had to be used. To ensure that our connectivity estimates were stable across sites and time, we scanned two individuals (travelling heads) over the course of three years on the three different scanners in which the imaging data for the actual studies were acquired. Both resting-state and diffusion weighted imaging scans were acquired using the exact same parameters that were used in the actual individual studies (see below).

Fifty-six ROIs selected from FreeSurfer's automatic parcellation/segmentation (Fig. S1A) were used to compute FC between all pairs of ROIs, as well as fractional anisotropy (FA) within each ROI. Test-retest reliability, estimated using Cronbach's alpha, was very high across all sessions, for each site, and for each travel head ($\alpha$s > 0.84, $p$s < .001; Fig. S1B, left). FC estimates expectedly showed greater variation than FA estimates, but correlations between sites were highly significant in both cases ($r$s > .60, $p_{FDRs}$ < .001; Fig. S1B, middle). In addition, FC and FA estimates for each ROI remained relatively stable across the different sessions (Fig. S1B, right). Similar results were obtained when using only the ROIs selected for the main study (Fig. S2).

For the high-resolution T1-weighted (MP-RAGE) structural images the following parameters were used: TR = 8 ms, TE = 4 ms, flip angle = 8°, voxel size = 1 × 1 x 1 mm³, FOV = 24 × 24 cm (studies S1,S2,S4,S5); TR = 2.53 ms, TE = 2 ms, flip angle = 7°, voxel size = 1 × 1 x 1 mm³, FOV = 19.2 × 25.6 cm (study S3); TR = 1.77 ms, TE = 3 ms, flip angle = 8°, voxel size = 1 × 1 x 1 mm³, FOV = 19.2 × 25.6 cm (study S6).

Whole-brain T2*-weighted images during rs-fMRI were acquired using a gradient echo, echo-planar imaging (EPI) sequence with the following parameters: TR = 2.5 s, TE = 30 ms, flip angle = 90°, voxel size = 3 × 3 x 3 mm³, FOV = 24 × 24 cm, 80 × 80 voxels, number of slices = 47, number of volumes = 190 (studies S1,S2,S4); TR = 1.43 s, TE = 30 ms, acceleration factor = 2, flip angle = 69°, voxel size = 3 × 3 x 3 mm³, FOV = 24 × 24 cm, 80 × 80 voxels, number of slices = 48, number of volumes = 190 (study S3); TR = 2.5 s, TE = 30 ms, flip angle = 90°, voxel size = 3 × 3 x 3 mm³, FOV = 28 × 31 cm, 92 × 92 voxels, number of slices = 46, number of volumes = 192 (study S6).

Diffusion-weighted images were acquired using the following parameters: TR = 9.5 s, TE = 88 ms, flip angle = 90°, voxel size = 2 × 2 x 2 mm³, FOV = 24 × 24 cm, 112 × 112 voxels, number of slices = 60, number of directions = 60 (b = 1000 s/mm²) (studies S1,S2,S4); TR = 5.5 s, TE = 114 ms, flip angle = 90°, voxel size = 1.6 × 1.6 ×1.6 mm³, FOV = 24 × 24 cm, 132 × 128 voxels, number of slices = 60, number of directions = 60 (b = 1000 s/mm²) (study S3); TR = 10.2 s, TE = 87 ms, flip angle = 90°, voxel size = 2 × 2 x 2 mm³, FOV = 60 × 60 cm, 120 × 120 voxels, number of slices = 70, number of directions = 60 (b = 1000 s/mm²) (study S6).

### Experimental paradigms

S1: The paradigm for this study consisted of four phases over two days: fear acquisition and fear reversal (day 1), followed by fear extinction (day 2). Participants viewed images of household appliances (16 CS in total), some paired with an electric shock (US). CS were presented for 1 s, embedded within 2 s video contexts that preceded them. The US, when delivered, lasted 0.75 s and followed the CS presentation. During fear acquisition, half of the CS were reinforced (CS+; 50% probability) whereas the other half were not (CS−). In fear reversal, contingencies were switched for half of the CS. US expectancy ratings were collected on each trial using a 4-point scale (2.5 s duration), and trials were separated by a fixation cross (7–9 s).

S2: In this study, participants viewed office scenes where a desk lamp's colour indicated CS type: one colour (CS+) was paired with a shock (US; 62.5% probability), while another (CS−) was not. Each trial included a fixation cross (6.8–9.5 s), a context image (1 s), and the CS

presentation (6 s). Fear acquisition involved 16 trials per CS type, followed by extinction and renewal phases without reinforcement.

S3: In this study, two geometric shapes served as CS, with one (CS + ) paired with a shock (US; 62.5% probability) during acquisition, while the other (CS−) remained unpaired. The experiment spanned two days: habituation (6 trials), acquisition (16 CS−, 16 CS+ , 10 CS+ US trials), and extinction (16 trials per CS type) occurred on day 1, while recall (not analysed) was on day 2. Trials lasted 8 s, with shocks (when present) delivered at 7.9 s. Inter-trial intervals varied between 14.3 s and 17.9 s.

S4: Participants learned to predict whether specific foods would cause a stomach ache based on context (restaurant). During acquisition (80 trials; 8 stimuli × 10 repetitions), each food was shown in one of two contexts for 3 s, followed by a question screen (max 4 s) and 2 s feedback. In extinction (80 trials), half of the stimuli were shown in the same context (AAA), half in a different one (ABA); extinction and distractor stimuli were included. The renewal phase (24 trials; 3 repetitions per stimulus) was conducted in the original context without feedback. Inter-trial intervals varied between 5–9 s.

S5: In this study, participants viewed three geometric shapes (CS + G, CS+ N, CS−) matched in luminescence and surface area. During acquisition (day 1), CS+ G and CS+ N were followed by an electrical shock (US; 62.5% probability), while CS− was never reinforced (8 trials). Each trial lasted 20 s: a jittered 0–2.5 s black screen, 8 s CS presentation, and 9.5–12 s inter-trial interval. During extinction (day 2), each CS was shown 8 times; CS+ N and CS− appeared in original size, while CS + G was presented across four sizes (100%, 75%, 50%, 25%, each size shown twice).

S6: This paradigm included two randomised-controlled studies examining the effects of systemic inflammation on fear learning. On day 1, participants underwent acquisition training, where three visual CS were paired with either visceral pain (CS+ US+vis), an aversive tone (CS+ US+aud), or no US (CS−). A total of 36 CS trials were presented (12 per CS type), with 75% reinforcement for CS+ (9 US+vis, 9 US+aud). CS were shown 6–10 s before US onset, and US lasted 14 s, with CS and US co-terminating. On day 2, extinction included the same CS sequence presented without reinforcement. Inter-stimulus intervals consisted of a fixation cross (8 s). Participants received either intravenous LPS or placebo 2 h before acquisition (study 1) or extinction (study 2). Only responses to CS+ US+vis were analysed.

For more details regarding the description of these studies see the supplementary section "Experimental paradigms (extended text)".

## Preprocessing of neuroimaging data

For the preprocessing of the resting-state fMRI data, fmriprep (version 20.1.1) was used, which included removal of the first two volumes, motion correction, slice timing correction and co-registration to the T1w image. The BOLD time-series were resampled onto native space (see also the section "Preprocessing (fmriprep boilerplate) in the Supplemental Methods").

For denoising, we extracted the expanded motion regressors from the fmriprep output (6 standard motion parameters, their quadratic terms and corresponding temporal derivatives; total of 24 regressors), in addition to the global signals and the mean signals within WM and CSF masks. In total, 36 regressors were used for denoising, as recommended by Satterthwaite and colleagues[75]. Next, we fitted a GLM using these sources of noise, and extracted the residuals of the resulting demeaned time series, which we then used for the functional/effective connectivity analysis described below.

FreeSurfer (v7.2) was run within fmriprep, thus, additionally providing the segmentation and parcellation maps (in native space) which were needed for the ROI extraction (see below).

For the preprocessing and analysis of diffusion-weighted imaging data, MRTrix3 (v3.0.4) and FSL (v6.0.6.1) were used. We initially ran the function dwidenoise from MRTrix3, which implements dMRI noise level estimation and denoising based on random matrix theory,

followed by mrdegibbs, which additionally removes Gibbs ringing artifacts.

Topup was then applied in order to estimate and correct susceptibility-induced distortions, followed by eddy-current correction, in order to correct eddy currents and movements in the diffusion data[76]. Finally, we ran FSL's tool eddy_quad for quality assessment of individual datasets.

## ROI extraction

Two different parcellation maps from FreeSurfer (Desikan-Killiany and Destrieux) were used for extracting regions-of-interest (ROIs).

A total of ten ROIs were extracted: amygdala (AMY), hippocampus (HIP), ventromedial prefrontal cortex (PFC), dorsal anterior cingulate cortex (ACC) and cerebellar nuclei (CEB), for both the left and right hemispheres. The choice of ROIs was determined by the SFB1280 consortium prior to any data acquisition (see also Fig. S36 and Table S1, for a literature review implicating these regions in learning and extinction). Note that for the dorsal anterior cingulate cortex and ventromedial prefrontal cortex, we chose the acronyms ACC and PFC, respectively, to be consistent with the other three-letter labels (AMY, HIP, CEB) and avoid excessive lettering in ROI pair labels in figures and tables.

AMY and HIP were taken from the automatic volumetric segmentation of the subcortical regions (aseg). We extracted the labels from the Desikan-Killiany atlas "medial-orbito frontal" and "caudal anterior cingulate", which correspond to the PFC and ACC, respectively (see Fig. 1B and Fig. S4).

We decided to include cerebellar nuclei as an ROI and not the cerebellar cortex, because the cerebellar nuclei are the sole output structure of the cerebellum (see[77] for a recent review). Given that FreeSurfer's automatic parcellation does not output the cerebellar nuclei (only the cerebellum as a whole), additional processing was required to construct the CEB ROIs. We used a pipeline similar to the one employed in our previous paper[78], which used the SUIT toolbox (https://www.diedrichsenlab.org/imaging/suit.htm). After aligning the T1w image of each individual to the ACPC, the cerebellum was cropped from the rest of the brain and normalised using DARTEL for the purpose of matching it to the SUIT atlas (in MNI space). We then applied an inverse normalisation to reslice the SUIT atlas into the functional space of each participant. Finally, the cerebellar nuclei (interposed, dentate and fastigial nuclei) were extracted and merged into one single ROI (Fig. S3).

All ROIs were resampled into functional (rs-fMRI) and FreeSurfer space. For both functional and effective connectivity, the ROIs in functional space were used. For structural connectivity, all ROIs were kept in FreeSurfer's native space, in order to take advantage of surface files (e.g., pial surface) that help improve the accuracy of tractography[79] (for examples of reconstructed tracts see Fig. S6). In addition, the PFC and ACC volumetric ROIs were also converted into surfaces (Fig. S4). These surface ROIs were only used for computing streamlines during probabilistic tractography, as tracking from surfaces from cortical brain regions has advantages relative to their volumetric counterparts[79].

We also extracted the bilateral thalamus using FreeSurfer's automatic segmentation. The thalamus of the contralateral hemisphere with respect to the seed/target CEB was used as a waypoint to more accurately guide tractography that included CEB ROIs (see Structural Connectivity section below for more details).

## Functional connectivity (FC)

Recently Mohanty et al.[80] tested the accuracy of several FC metrics by evaluating a support vector machine classifier using a neighbourhood component analysis feature selection. They found that FC was better characterised by a combination of nine different but complementary metrics (a composite metric) than any metric alone. The authors

reasoned that Pearson correlations - the most common measure of FC - only look for statistical linear time-dependencies, whereas there could still be underlying statistical dependencies between BOLD signals that are poorly captured by Pearson correlations (e.g., non-linear dependencies).

Therefore, for the present study, we followed Mohanty's recommendation and computed a total of nine functional connectivity metrics: Pearson correlation, cross-correlation, dynamic time warping, Euclidean distance, Manhattan distance, Wasserstein distance, mutual information, coherence and wavelet coherence (see Supplemental Methods, Table S4, for the mathematical implementation, Fig. S7 for correlations among these metrics, and Fig. S8-S10 for comparisons among the different FC metrics within our network).

Note that we also repeated all analyses using Pearson correlations as the FC metric in all our models. Results were largely consistent, with very similar connections in the analyses using the composite metric and Pearson correlations. In addition, the regression estimates from the two approaches were strongly correlated ($r = .60$, $p < .001$; Fig. S38), indicating that our findings are not critically dependent on the choice of FC metric. Nevertheless, while the composite metric may provide a broader coverage of FC, we acknowledge that even this metric may not fully capture the complexity of FC dynamics, and future work will be needed to refine these metrics further.

For the FC analysis within the core learning network, a multilevel model was fit with the learning estimates as the outcome variable and the connectivity values for the 20 ROI pairs as the predictors of interest. Age and sex were included as covariates, and participant nested within study was treated as a random effect. Correction for multiple comparisons between the different connections was performed using the false discovery rate (FDR) method.

## Effective connectivity (EC)

EC was estimated using spectral dynamic causal modelling (spDCM), a toolbox for SPM12[31] (Matlab v2022b). spDCM is a variant of standard DCM that is especially developed for modelling resting-state data. It is based on the cross (power) spectral density of the observed BOLD time-series (see the Supplemental Methods for a comprehensive mathematical description of spDCM; see also Fig. S10 for the correlation between EC estimates and those from the FC metrics coherence and cross-correlation).

From the preprocessed and denoised functional images, a single representative time series was computed for each ROI by performing a principal component analysis across voxels and retaining the principal eigenvariate. This procedure has the advantage that it is more robust to outliers when computing EC.

The DCM model was then specified for each participant. Because we were interested in ipsilateral connections (with the exception of the contralateral connections involving the cerebellum), we set a prior with zero mean and zero variance for those connections that we wanted to exclude from our models. Estimates of parameter uncertainty were computed by extracting the diagonal of the covariance matrices for each individual connection (these estimates were used in the EC LASSO models described below).

After spDCM was computed for each subject individually, a structure was returned with the values in the matrix containing the mean value of the Gaussian distribution for the connections of interest (commonly known as A-matrix). These mean values comprise the connectivity parameters that indicate the individual-level effective connectivity from one brain region to another brain region.

Subsequently, we computed a group-level DCM by assembling all subject-level DCMs and providing them asf input to the group DCM analysis using the framework of Parametric Empirical Bayes (PEB). PEB takes into account the estimated covariance between parameters and subjects in a random effect analysis (see Supplemental Methods for technical details about PEB). Only extrinsic connections with posterior probabilities above 95% (which is equivalent to a log Bayes factor of 3) were considered, which corresponds to "strong evidence" within a Bayesian framework.

Note that we use the terms "excitatory" and "inhibitory" in the context of EC to denote influences that increased (positive sign) or decreased (negative sign) activity in target regions, respectively. Even though the excitatory/inhibitory labelling in the present study does not refer to cellular or synaptic mechanisms – as we have not directly measured neurotransmitter action – we, nevertheless, use these labels in the text to be consistent with the terminology used in other DCM-fMRI studies[31].

## Structural connectivity (SC)

After preprocessing the diffusion data, we proceeded with the tractography analysis using FSL. First, we fitted a diffusion tensor model at each voxel, which returned the first three eigenvectors and corresponding eigenvalues, as well as a fractional anisotropy (FA) map.

Because all subsequent analyses were done in diffusion space, we used the FA map to create all necessary transformation matrices from diffusion (dwi) to anatomical (T1w) and FreeSurfer spaces. The reason for using the FA image was that it had a more similar contrast to the anatomical image than non-weighted diffusion (B0) volumes, so it provided a slightly more accurate registration. All transformation matrices (diffusion -> FreeSurfer, FreeSurfer -> diffusion, diffusion -> T1w, T1w -> diffusion) were computed using boundary-based registration (BBR).

Next, we ran an analysis using Bayesian Estimation of Diffusion Parameters Obtained using Sampling Techniques (BEDPOSTX), which models crossing fibres within each voxel of the raw diffusion data, and creates distributions of diffusion parameters at each voxel.

Probabilistic tractography was then estimated using the samples from the distributions above. Note that tractography was run in diffusion space, but the transformation matrix diffusion -> FreeSurfer was provided, since all ROIs were in FreeSurfer space and the pial surface was used as a stopping mask.

For the tracking of fibres from cortical regions (i.e., ACC and PFC) we used surfaces instead of volumetric ROIs, as it has been suggested that seeding from surfaces is superior for the cortex[79]. However, for subcortical regions and cerebellar nuclei, we used the volumetric ROIs, since these regions tend to contain a high degree of anisotropy.

As a means to guide tractography, we included the pial mask as a stopping mask, which effectively prevented tracts from crossing the grey-white matter interface (which would be biologically implausible). Furthermore, for any seed regions, we further restricted tractography to exclude streamlines that did not stop by the target region. In other words, tracts were terminated and included in the total streamline count if and only if they reached the target region from the seed region. Finally, we also constrained tracts to bypass any regions containing non-white matter voxels (e.g., CSF, skull).

One limitation with the approach described above is that it can only be meaningfully used for ipsilateral connections, as including the grey-white matter interface will inevitably discard any streamlines that attempt to cross hemispheres. This is particularly problematic for connections involving the cerebellum, since anatomical connections of the cerebellar hemispheres involve to a large extent the contralateral cerebral hemisphere with cerebellar output crossing at the level of the brainstem[81]. In order to provide a similar degree of tractography accuracy for the cerebellar connections, we restricted the movement of the tracts to/from the cerebellar nuclei by (1) defining two stop masks – the pial surface ipsilateral to the cerebellar seed, and the cerebellar hemisphere, and (2) using the thalamus as a first waypoint, such that only tracts that initially run via the thalamus en route to the target ROI were considered valid streamlines. The rationale for point (2) is that the output of the cerebellar nuclei is connected with various cortical areas via the thalamus[81].

#### Learning measures

*SCR measures.* For analysis of the SCR data we used the Matlab toolbox Psychophysiological Modelling (PsPM, version 6.0.6)[82]. Similar to analysis of fMRI data, PsPM creates an explicit mathematical "forward" model of the data-generating process with unknown parameters. This model is then inverted to yield the most likely parameter values, given the data. In the present case, we estimated the amplitudes of centrally generated CS− and US-related sudomotor responses on each trial, which serve as a learning index.

The raw data for each subject were initially trimmed to the task duration and, subsequently, filtered using an adaptive filtering algorithm with varying numbers of time points, such that an optimal number could be determined that maximally reduced gradient artefacts. Subsequently, residual artifacts in the filtered data were detected using an automated quality assessment procedure[83]. Finally, each dataset was visually checked for residual artifacts that may have been missed by the automated artifact detection tool, which were marked by one of three researchers with extensive training in preprocessing SCR data. Any detected artefact periods were logged and later ignored during data analysis. Participants with an excessive amount of artifacts were excluded from further statistical analysis, which was agreed by the three researchers, who were blinded to the results of the actual analysis. Note that most of these exclusions were the result of failed recordings due to technical difficulties with the equipment or sudden abortion of the scanning session. In total, 56 acquisition and 79 extinction datasets were excluded due to flat or coarse SCR data (see Fig. S13 for examples of typical excluded datasets).

After preprocessing, data for all but one study (S6) were analysed with a standard non-linear model[84] using a canonical skin conductance response function[85] (see also Supplemental Methods for a mathematical description of this model). This approach is appropriate for the CS−US duration used in these studies (around 7 s). For all phases (acquisition, extinction and renewal), we modelled the following events: Context (fixed latency), CS onset (fixed latency), CS interval (flexible latency, sudomotor burst dispersion fixed at 0.3 s) and US (fixed latency).

In order to avoid bias, PsPM assumes the same event sequence for all trials, so in the case of non-reinforced trials (no US), we modelled US omissions by adding an event of the same duration as the actual US in the place where the shock would have occurred during reinforced trials.

To exclude the possibility that the estimated response to the CS could have been confounded by the response to the US (due to the overlap of the elicited SCR), we applied two additional steps. First, we gradually decreased the modelled time interval between CS onset and US onset, to the point that there would be no differences between CS + US+ and CS + US-. Second, we ensured post hoc that the estimated sudomotor impulse did not overlap in time with the onset of the US. Both these procedures ensured that the response to the CS during reinforced trials was unlikely to have been contaminated by the response to the US. In consequence, there was no evidence for a difference between CS + US+ and CS + US- ($t(575) = -.52$, $p = .61$, $d = .04$, 95% CIs = $[-.16, .09]$), suggesting that our method was not biased by the US presence.

For the acquisition phase of S6, in which the US duration was very long (14 seconds), we used a different approach. Crucially, the standard non-linear model requires assumptions on the number and distribution of anticipatory responses during the CS−US time window, and these have not been thoroughly tested and validated in long-window paradigms. Hence, we opted for a method that dispenses with these assumptions and has been successfully used to estimate spontaneous fluctuations which can occur any time[86]. Specifically, we modelled one response per two seconds and estimated its amplitude and onset, across the entire acquisition. We then assigned the estimated sudomotor response to the experimental events that occurred

at the same time. Next, we summed the estimated amplitude of all responses occurring during each CS, and divided them by the CS duration, thus providing a score of the anticipatory sudomotor activity per CS.

Once the modelling was completed for all studies, the results for each trial were manually inspected to ensure that a proper model fit was attained. Finally, for each subject, we extracted the amplitude for each trial and for each phase of the experiment. These data were subsequently analysed using R (version 4.2.2; https://www.r-project.org/).

Using each study's parameter estimates, we then selected the conditions of interest that we wished to model (see Table S3). Some studies (e.g., S5) contained fewer than four non-reinforced CSs in their experimental design, which did not allow for an accurate assessment of learning across time. Therefore, we combined reinforced trials (CS + US + ) and non-reinforced trials (CS + US-) in our modelling. As noted above, we ensured the amplitude differences between CS + US+ and CS + US- were indistinguishable and were not significant in each single study. Further tests confirmed that the response amplitudes for CS + US+ were not contaminated by subsequent US responses (see Fig. S14).

For generating a single participant-specific learning score that could characterise individual learning performance, we employed the following sequential steps (see Fig. 3B for a visual example):
1) A theory-free polynomial regression of the 2nd order was conducted on the amplitude scores extracted from PsPM as the dependent variable; trial number, CS type and their interaction were used as predictors of interest.
2) After the model fit, the model predictions were extracted for each participant.
3) The difference in the predictions between each trial and the previous trial was calculated for each CS type separately and summed into a single score.

Thus, each participant's learning was characterised by two scores (one for CS+ and one for CS−). The difference between the CS scores (CS+ minus CS−) gave us an indication of the learning rate relative to the baseline.

#### Behavioural responses

The behavioural data in S4 consisted of binary decisions (whether a food item gave stomach ache) as the dependent variable (see above for the description of the paradigm used in S4) and trial number as a predictor of interest.

Because we were interested in individual rates of learning, we ran a multi-level logistic regression analysis using a random intercept for each participant and a random slope for trial number (see Fig. S37 for the correlation between our modelled responses and the original averaged scores reported by Lissek et al.[21–23,65]). The extracted coefficients included, for each participant, an intercept, representing the average learning, as well as a slope for trial number, representing the learning rate.

However, the slopes derived from logistic regression models cannot be interpreted on their own, since we require at least two parameters (i.e., $\beta_0$ and $\beta_{time}$) to determine the shape of the logistic curve. Specifically, $\beta_0$ indicates the general ability of the subject, whereas $\beta_{time}$ indicates the subject's ability to learn over time without considering the overall skill.

Therefore, we computed the probability of success for each participant and trial, and subsequently calculated the expected number of correct trials after 8 attempts (since there were always 8 unique trial types in each experiment) based on these probabilities. Scores closer to 8 would mean that participants learned successfully, whereas scores closer to 4 would mean that participants were at chance levels. These scores were thus used as a proxy for how well each individual learned through time (see Fig. S15 for a distribution of these scores).

This procedure was used for the analysis of acquisition, extinction and renewal.

**Predicting individual differences.** The main goal of the present study was to predict the efficacy of learning and extinction by multimodal brain connectivity patterns. This purpose required three learning parameters per participant (one for acquisition, one for extinction and one for renewal) in each and every study included in the present research.

Different studies utilised different experimental paradigms, different numbers of trials/conditions, and different dependent variables. To reduce this inhomogeneity, we used a modelling strategy such that the different types of models we implemented for the SCR and behavioural studies resulted in similar learning parameters (i.e., a single score per subject that reflected learning *across time* during acquisition, extinction and renewal).

Furthermore, we standardised the learning estimates on a study-by-experiment level, such that each study/experiment had an average estimate of 0 and variance of 1. This step was important because it brought all study-experiment combinations into a common unit of measurement, while preserving the relative distances between individuals within that study-experiment combination.

An additional advantage of this procedure is that it effectively helps to reduce differences across studies due to unwanted site effects (e.g., scanner, sample size, etc.). To explore how much non-trivial variation was present after standardisation, we ran a simple nested multilevel model using participant, study and experiment as separate random effects and the standardised learning variable as the dependent variable. The variance estimates of the random effects "study" and "experiment" were zero (Table S5), indicating a degenerate model (i.e., the between-study and between-experiment variability is insufficient to justify their inclusion as random effects, over and above the participant random effect). Because all models above were fitted using maximum likelihood, we could compare models with and without the random effects of study and/or experiment. The results of these model comparisons indicated equivalent AICs to using participant as the sole random factor (Table S6). In addition, a ComBat-corrected model showed no improvement in fit relative to the original model (comparable AIC), indicating negligible site effects (Table S7).

LASSO regression models were then built using learning score as the dependent variable, and either FC, EC, or SC values as predictors of interest (i.e., in three separate models). Age and sex were used as covariates in all models. For models based on FC and SC, we included 20 connectivity predictors corresponding to the 20 possible undirected connections among the 10 ROIs (with 5 ROIs in each hemisphere and only within-hemisphere connections). For the EC models, there were a total of 40 connectivity predictors (A→B and B→A are treated as separate connections). Importantly, all models were independently estimated within a regularisation framework, which penalises model complexity and shrinks uninformative predictors towards zero[87]. This procedure effectively accounts for differences in the number of predictors and reduces the risk of overfitting.

Regularised regression methods such as LASSO can be difficult to interpret in the presence of multicollinearity since they will discard almost arbitrarily one of the collinear predictors. Thus, before each LASSO model, we ran a multiple regression analysis in order to assess several multicollinearity diagnostics, which included the variance inflation factor (VIF), tolerance, eigenvalues, condition indices and variance proportions. None of the models showed a sufficiently high degree of multicollinearity that would warrant further investigation[88] (see Fig. S17).

Each model was built using either all studies or a subset (see Table S8). To ensure robust and generalisable results, we used a three-step nested cross-validation procedure. First, LASSO was run with a k-fold cross-validation procedure, separately for acquisition,

extinction, and renewal. The cross-validation function initially randomly split the data into 10 equal folds. The model was subsequently trained on nine folds and tested on the one held-out (test) fold. This process was repeated for all 10 folds, ensuring that each fold served as the test set once. We then extracted the average mean squared error (MSE) across these 10 different splits as a measure of how well the model generated from the training data can predict the test data. Please note that in this procedure, the feature selection (i.e., identification of predictive connections) was exclusively done in the training set (90% of the data) and then these features were used to predict data in the test set.

The procedure above uses a particular set of folds, meaning that the effects could be specific to those folds. Thus, in the second step, we included an outer loop of cross-validation to ensure the stability of our LASSO results. Specifically, we applied the 10-fold cross-validation procedure described in step one above one hundred times, each time randomly partitioning the data into the 10 folds.

Finally, for each of the 1000 folds (10 × 100 outer-loop iterations), we supplied the cross-validation function with a decreasing sequence of values based on the following formulas:

$$\lambda_{\max} = \frac{j=1,\,...,\,p^{\max} \left| \sum_{i=1}^{n} x_{ij} y_i \right|}{n},$$

$$\lambda_{\text{path}} = \left\{ \lambda_k \,\middle|\, \lambda_k = \exp\left[ \log(\lambda_{\max}) + \frac{k}{K-1}\left(\log(\lambda_{\max}\epsilon) - \log(\lambda_{\max})\right) \right], k = 0, 1, \dots, K-1 \right\}$$

Here, $x \in R^{n \times p}$ is the design matrix, $y \in R^n$ the response vector, $n$ is the number of samples, and $p$ the number of predictors. The lambda path contains $K$ values equally spaced on a log scale from $\lambda_{\max}$ to $\lambda_{\max} \cdot \epsilon$ with $\epsilon = 10^{-4}$. This custom manual sequence of $\lambda$ values (i.e., $\lambda_{\max}$) maximised the chances of finding the optimal regularisation parameter $\lambda$ within each fold, controlled the regularisation range and provided numerical stability.

In summary, for each of the 10 folds in each of the 100 cross-validations, we tested 1000 models with 1000 different regularization parameters $\lambda$, given by $\lambda_{\max}$ computed above. The repeated cross-validation iterations identified features that were consistently selected across different splits of the data, ensuring that the connections presented in Fig. 4 reflect the most stable and reliable patterns identified by the LASSO model. Finally, to extract significant connections, we counted the number of times each of the connections was chosen in 1000 LASSO models (with the respective optimal regularization parameter $\lambda$) and then we applied a binomial test to identify connections with a significant number of contributions (see stars in Fig. S22).

The calculation of significance in regularised regression is controversial and an area of much debate, as p-values and confidence intervals in LASSO models are biased due to the regularisation process, which is conducted to reduce variance in the parameter estimates[89]. Alternative methods to compute p-values have been suggested, but none have been found robust. In the present study, we defined the nonzero coefficients from LASSO, obtained via the cross-validation procedure described above, as the "significant" variables under the penalised optimisation framework L1. Nevertheless, we also provide standard p-values for each of the relevant connections from the LASSO models as a comparison. LASSO was run on one hundred random samples of observations, counting the nonzero connections and testing each connection using binomial tests while correcting for multiple comparisons (see Fig. S22).

Overall model performance was assessed for each experimental phase separately (acquisition, extinction and renewal). First, the data were split in half, such that one half was used as the training set and the other half as the testing set. Surrogate datasets were constructed by shuffling the learning-connectivity correspondences of the testing sets across subjects. The mean-squared errors (MSEs) were computed for the original and surrogate models, and the difference was taken as a measure of performance. Permutation tests were performed by

comparing the original difference against the null distribution, which was constructed by randomly multiplying the MSE differences by either −1 or 1 ten thousand times (see Tables S11-S13).

For EC models, we also used the previously computed estimates of variability for each connection (see above) as weights during model fit, so as to mimic the group-level PEB that also uses this information.

Finally, in order to ensure our results were not due to the specific statistical LASSO method we used, we re-ran all of our models using standard multiple regression, ridge regression and elastic net. The LASSO coefficients were well within the range of the coefficients provided by these alternate methods, and there were very robust positive correlations among the coefficients from the different methods (Fig. S18).

### Generalisability analysis

After fitting our LASSO models, we examined the generalisability of these models by running four different types of analyses.

First, we grouped studies into two larger groups: a fear learning (FL; S1,S2,S3,S5,S6) and a predictive learning (PL; S4) group. We applied Monte-Carlo cross-validation to extract mean-squared errors (MSEs) of different combinations of the testing set participants. Next, we trained a LASSO model on all data from one paradigm (e.g., fear learning; using the same lambda path to compute the optimal lambda, as in the main LASSO model described above) and applied it to randomly selected 50% of data from the other paradigm (e.g., predictive learning). This was repeated 100 times, and each time we selected a different random subsample of 50% of the test data. The training data were always the same across the 100 repetitions. As a result of the 100 repetitions, we obtained a distribution of MSE values (blue error bars in Fig. 5A-C).

Subsequently, we created surrogate datasets by randomly shuffling the relationship between learning and connectivity estimates in the test (unseen) data. Again, we selected 50% of these surrogate test and computed how well the model (based on the actual data in the training set) could predict these test data, resulting in a surrogate MSE value. This was repeated 100 times with different shuffles of the test data (black error bars in Fig. 5A-C).

Then, the difference between the original and surrogate MSEs was used as a performance measure. Statistical significance was then assessed using non-parametric permutation tests. Specifically, the null distribution was constructed by randomly multiplying each of the 100 MSE differences by either −1 or 1 with equal probability and again averaging these differences. Next, we recalculated the mean difference across 10,000 permutations, representing the expected variation if the observed differences were due to chance. Finally, we compared the observed mean difference to this null distribution and calculated a two-tailed p-value, which reflects the proportion of permuted means that were as extreme as or more extreme than the observed difference.

In the second generalisability analysis, we employed a leave-one-group-out (LOGO) cross-validation by splitting the data such that each training set comprised of all studies except one (e.g., S2–S6, but not S1), and the left-out study (e.g., S1) was used as the test data. Every study was used as the test dataset once. A multiple regression was fit on the training data and the model tested on the testing data. The Pearson correlation coefficient for the averaged predicted and observed values served as an index of generalisation. Note that, in this analysis, we aimed to test if the set of connections identified in the previous LASSO models (shown in Fig. 4) generalised across the dataset, and thus it would not have been appropriate to perform feature selection here.

In the third generalisability analysis, we enquired whether our model predictions could be useful to ascertain task-based FC, as the model for the acquisition phase revealed significant functional connections with respect to the entire sample. For that purpose, we selected one fear conditioning study for which task fMRI data were available (S2, two experiments, 152 participants) and computed FC during both CS+ and CS− trials using all FC metrics described above and for all combinations of the relevant ROIs.

A multiple linear regression model was constructed using the significant connections from the overall LASSO model, and the model p-value was contrasted with one from a model that contained randomly selected connections. As the vast majority of FC studies computed Pearson correlation coefficients as a proxy for connectivity, our FC predictors were also based on Pearson correlations.

Finally, to extrapolate our findings to potential new data, we also ran two types of analyses: simulations and bootstrapping. For the simulations, we drew samples from a multivariate normal distribution since the outcome variable of learning was normally distributed (Fig. S19). The covariance matrices for each group (PL and FL) were computed for all numerical predictors and covariates, to account for correlations among these variables. To accommodate potential variations around the sample mean and covariance (as we would not expect the sample mean and covariance matrix of subsequent studies to equal exactly the empirical estimates), we allowed some degree of sampling error around our estimates by unscaling the variables. For the bootstrapping, we performed the same analysis as above but using bootstrap samples once with replacement and once without replacement.

Two multiple regression models were computed (one for FL and another for PL studies), using learning as the outcome variable, the connectivity pairs from the selected LASSO models as predictors and sex and age as covariates. One hundred independent datasets were generated, each contained 40 observations (we tested various other sample sizes [90, 300, 1000], but the results were identical; see Fig. S20). For each dataset, r-squared from a multiple regression was computed for the actual LASSO predictors, as well as for pseudo-random predictors that were not present in the actual LASSO model but consisted of regions from the core learning network.

Importantly, we selected the exact same number of pseudo-random connections as the LASSO model within each specific type of connectivity (e.g., if the LASSO model for effective connectivity contained 3 connections, the random model for the effective connectivity would also contain 3 connections not present in the LASSO model).

We also computed connectivity within the lateral and fourth ventricle, as we did not expect these regions to predict learning, and, thus, functioned as an additional baseline.

### Reporting summary

Further information on research design is available in the Nature Portfolio Reporting Summary linked to this article.

## Data availability

Public sharing of the raw data is not permitted because participant consent and ethics approvals do not allow unrestricted public data sharing, and the datasets are subject to institutional and legal data-sharing agreements across the contributing institutions. Raw data are available from the corresponding author upon reasonable request and subject to approval by the contributing institutions and a data-sharing agreement where required. Access will be granted to qualified researchers for non-commercial scientific research purposes; decisions are typically made within 4–6 weeks. Source data are provided with this paper.

## Code availability

The code for data preprocessing and analysis is available at https://github.com/caadgomes/extinction-learning-study.

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

## Acknowledgements
This study was supported by the DFG SFB 1280 "Extinction Learning" (Project Nr. 316803389, F02: N.A., T.S., D.T., R.K., C.A.G.). We thank Bianca Hagedorn for her help in data collection, Cosima Clotten, Julia Stengel, Esther Yadgarova, Fabian Wissing and Lina Schmidt for their help with the PsPM analysis and figures, and Aarti Swaminathan for her help with some sections of the discussion.

## Author contributions
C.A.G., G.B., T.M.E., M.C.F., C.F., A.K., F.L., D.M., A.N., R.J.P., A.T. collected the data. S.E., H.E., E.G., S.L., C.J.M., M.T., O.T.W., O.G., H.H.Q., D.T., N.A. devised the experiments. C.A.G., D.R.B., A.R. and T.S. designed and performed the analyses. R.K., D.T., T.S. and N.A. supervised this project. C.A.G. and N.A. wrote the manuscript. C.A.G, D.R.B., A.R., G.B., S.E., H.E., T.M.E., C.F., E.G., F.L., S.L., C.J.M., D.M., A.N., R.J.P., J.E.S., A.T., O.G., H.H.Q., R.K., D.T., T.S., N.A. commented on the manuscript.

## Funding

## Competing interests
The authors declare no competing interests.

## Additional information

[1]Department of Neuropsychology, Ruhr University Bochum, Bochum, Germany. [2]Department of Neurology, University Hospital Essen, Essen, Germany. [3]Center for Translational Neuro- and Behavioral Sciences, University Hospital Essen, University of Duisburg-Essen, Essen, Germany. [4]Erwin L. Hahn Institute for Magnetic Resonance Imaging, University of Duisburg-Essen, Essen, Germany. [5]Department of Imaging Neuroscience, UCL Queen Square Institute of Neurology, University College London, London, UK. [6]University of Bonn, Transdisciplinary Research Area "Life & Health", Centre for Artificial Intelligence and Neuroscience, Bonn, Germany. [7]Turner Institute for Brain and Mental Health, School of Psychological Sciences, Monash University, Clayton, Australia. [8]Monash Biomedical Imaging, Monash University, Clayton, Australia. [9]CIFAR Azrieli Global Scholars Program, CIFAR, Toronto, Canada. [10]Department of Medical Psychology & Medical Sociology, Ruhr University Bochum, Bochum, Germany. [11]Institute of Medical Psychology and Behavioral Immunobiology, University Hospital Essen, Essen, Germany. [12]Department of Psychology and Neurosciences, Leibniz Research Centre for Working Environment and Human Factors at the Technical University of Dortmund (IfADo), Dortmund, Germany. [13]Department of Neurology, BG University Hospital Bergmannsheil, Ruhr University Bochum, Bochum, Germany. [14]Department of Cognitive Psychology, Ruhr University Bochum, Bochum, Germany. [15]Institute of Psychology, Department of Educational Sciences and Psychology, TU Dortmund University, Dortmund, Germany. [16]Department of Psychiatry and Psychotherapy, Center for Mind, Brain and Behavior, Philipps-Universität Marburg, Marburg, Germany. [17]Genetic Psychology Lab, Ruhr University Bochum, Bochum, Germany. [18]Department of Biopsychology, Ruhr University Bochum, Bochum, Germany. [19]Research Center One Health Ruhr, Bochum, Germany. [20]High Field and Hybrid MR Imaging, University Hospital Essen, Essen, Germany. ✉e-mail: carlos.assuncaodias@rub.de

