## [Transparent Peer Review file · Nature Communications]

Predicting individual differences of fear and cognitive learning and extinction

Corresponding Author: Dr Carlos Alexandre Gomes

Version 0:

Reviewer comments:

Reviewer #1

(Remarks to the Author)

This paper presents the effects of fear learning in the brain. Specifically, the authors found a distinct role of structural, functional and effective connectivity in predicting fear renewal, learning and extinction, respectively. The work analyzed 500 participants with a solid framework and control analyses. The paper is valuable and can be published in Nature Communications. However, I found that a solid reshaping of the text is mandatory and several concerns need to be addressed.

The authors start by mentioning the harmonization of the experiment without introducing them and without specifying how this harmonization has been performed. Please rephrase the entire paragraph by introducing the experiments and then describing the harmonization.

A brief description of the experiments is not mentioned before the methods part and without reading the caption of Figure 1.

In addition, in the description of the experiments, some details are unclear to me. For example, is the acquisition phase included in every experiment? In the experiment S4 it is not clear if the stomachache presentation is the same as the US. The suggestion is to provide a detailed but concise description of the experiments in the text in order to allow the reader to not go up and down to find useful information.

Moreover, some acronyms (e.g. CS+, CS-) are mentioned before defining them, please read carefully the manuscript and revise it in depth.

It is not straightforward to assess how many parameters were included for each prediction model. To my understanding EC has more parameters than FC and SC. How this difference may impact the results?

I am a bit concerned regarding the composition of the FC metrics, given that they looked at time course similarity but from different perspectives. This should be carefully interpreted and validated. For example, classical correlation analysis, has the same behaviour of the proposed approach or is it specific to the set of metrics used?

Renewal is included in two studies. Do the differences of extinction and acquisition hold for two studies as well?

You delivered an electrical stimulus to the skin. Since I am not an expert of this type of stimulation. Are there any possible confounds in the brain responses of this stimulation? If yes, how did you control for these confounds?

The renewal behavioural results look alike. I mean that I would expect that CS+ has an increase rather than a decrease. What am I missing?

The choice of the ROIs is an a-priori choice, motivated by literature, but to validate the findings the authors should test whether and how this choice is specific to this network or can be found with other ROIs.

In Figure 3B, results are a bit counterintuitive. I mean that during FL you found no important connections while when including S4, FC connectivity results to be important. However, when using S4 alone FC is still important. This looks weird

or at least should be discussed and analyzed more in depth to understand how this has happened.

The authors mentioned that "Note that it is possible for a connection to be identified as significant when data are pooled across all studies, even it was not significant in any individual study". Can you elaborate on this? I mean this is crucial for establishing the validity of your results, can you provide a clear example or simulation that validates your sentence.

In Figure 4J-L, it is not clear what the colors are indicating.

How did you find that some EC connections are inhibitory and others are excitatory? I mean, as far as I know, EC gives you the direction of the connections so if a region is influenced or influences another. While the excitatory and inhibitory mechanisms are more an interpretation of the directionality. I would suggest reframing the discussion to assess for the intrinsic limitation of DCM analyses.

(Remarks on code availability)

I would recommend to add the versions of software used in a requirements.txt or project.toml file.
It is not super-reproducible but can be improved.

Reviewer #2

(Remarks to the Author)

I appreciate the authors for addressing my comments. I donot have other concerns.

(Remarks on code availability)

Version 1:

Reviewer comments:

Reviewer #1

(Remarks to the Author)

The authors carefully addressed all my concerns.

(Remarks on code availability)

made.

Response to Reviewer #1's comments

We thank the reviewer for the insightful comments on our manuscript. We have now revised our manuscript accordingly, performing additional control analyses and updating the text. We believe that these changes have greatly improved the clarity and strength of our manuscript. Please see below our point-by-point response to each of the reviewer's comments.

Comment 1: The authors start by mentioning the harmonization of the experiment without introducing them and without specifying how this harmonization has been performed. Please rephrase the entire paragraph by introducing the experiments and then describing the harmonization.

Response: Thank you for this comment. Please note that the harmonisation paragraph at the beginning of the Results section refers to the harmonisation of the scanning sequences, not of the experimental paradigms. Accordingly, the results in Fig. S1-S2 show resting-state and diffusion-related connectivity without the administration of an experimental task.

But we agree that, as we have written it, this is not entirely clear, so we have updated the first paragraph of the Results section which now reads:

"Because participants were scanned at three different locations using three different 3T MRI systems from two vendors, we carefully harmonised and optimised scanning sequences across centres. Specifically, we tested the stability of our resting-state and diffusion acquisition protocols by scanning two individuals over the course of three years on all three scanners using the exact imaging parameters of the actual studies. FC and SC measures demonstrated high test–retest reliability, both when using whole-brain ROIs and when restricting the analyses to the study-specific ROIs (all Cronbach's alpha > .80; Fig. S1–2)."

In this section, we also briefly introduce the experiments (please see response to next comment).

Comment 2: A brief description of the experiments is not mentioned before the methods part and without reading the caption of Figure 1.

Response: We have now added a brief description of the experiments after the harmonisation paragraph. It now reads:

"We then acquired rs-fMRI and DWI data from a large group of participants (rs-fMRI: N=509; DWI: N=463) who took part in either a fear learning (FL; studies S1, S2, S3, S5 and S6) or cognitive predictive learning (PL; study S4) experiment (Fig. 1C; see also Methods and Supplemental Methods: Experimental paradigms for more details).

The FL studies employed variations of classical fear conditioning, in which some visual CS were partially reinforced by an aversive US (CS+; electrical shocks in S1–S3 and S5, visceral stimulation in S6), whereas other stimuli were never reinforced (CS-). The PL studies differed in that participants learned predictive associations between food items and a putative outcome (“stomach ache”) in different contexts. All paradigms included acquisition and extinction phases, the latter consisting of unreinforced CS trials in a different context. Renewal was tested in S2 and S4 by showing the same stimuli in the acquisition context.”

Comment 3: In addition, in the description of the experiments, some details are unclear to me. For example, is the acquisition phase included in every experiment? In the experiment S4 it is not clear if the stomachache presentation is the same as the US. The suggestion is to provide a detailed but concise description of the experiments in the text in order to allow the reader to not go up and down to find useful information.

Response: We have added a description of the studies at the beginning of the Results section (see response to comment 2 above). Regarding the specific questions in this comment:

- 1) Yes, the acquisition (and extinction) phase was included in every experiment. However, only S2 and S4 had a renewal phase.
- 2) Usually, a US elicits an unconditioned response. Most fear learning studies use electric shocks as the US, since they reliably produce a physiological and aversive reaction. In S4, the “stomach ache” is not actually delivered as a physical stimulus. Participants pressed a button to indicate whether the image of a particular food was followed by a sentence stating that this food image caused a stomach ache or not, so it should be thought more of a cognitive expectation task than an aversive experience. Thus, the outcome is predictive (hence the label cognitive predictive task), rather than a direct US that evokes a physiological response.

Both points were incorporated in the text above (see comment 2).

Comment 4: Moreover, some acronyms (e.g. CS+, CS-) are mentioned before defining them, please read carefully the manuscript and revise it in depth.

Response: We apologise for this oversight. We have now rectified all acronyms throughout the text.

Comment: It is not straightforward to assess how many parameters were included for each prediction model. To my understanding EC has more parameters than FC and SC. How this difference may impact the results?

Response: We apologise again for the lack of clarity. The number of parameters in FC, SC and EC are now reported on page 32/33. In short, with 5 ROIs in each hemisphere there are 10 possible undirected connections (e.g., both HIP-AMY and AMY-HIP count as one connection) for both FC and SC in each hemisphere. For the two hemispheres, there are thus 20 connectivity-related predictors in each of these models. For EC, the connection from A to B is different from B to A, so there are 40 connectivity-related predictors in EC.

Regarding whether differences in the number of parameters in the model could affect the results, we believe it has negligible to no impact. Note that all models were regularised, which penalises model complexity and reduces overfitting (Friedrich et al., 2023; Hastie et al., 2009), effectively accounting for the larger number of predictors in EC. Predictors which do not contribute substantially will be shrunk towards zero in regularised models, thus reducing variance inflation. Furthermore, our procedure tuned model complexity by careful selection of the penalty parameter (i.e., lambda; Tibshirani, R., 1996). So, by applying this regularisation framework (balancing model fit with complexity), we ensured that we do not increase the risk of overfitting by having more predictors. Also, note that we analysed FC, SC and EC independently, that is, the predictors in one model did not influence those in the other models.

We stressed both these points on page 32/33, where we now write:

“For models based on FC and SC, we included 20 connectivity predictors corresponding to the 20 possible undirected connections among the 10 ROIs (with 5 ROIs in each hemisphere and only within-hemisphere connections). For the EC models, there were a total of 40 connectivity predictors ($A \rightarrow B$ and $B \rightarrow A$ are treated as separate connections). Importantly, all models were independently estimated within a regularisation framework, which penalises model complexity and shrinks uninformative predictors towards zero⁸⁶. This procedure effectively accounts for differences in the number of predictors and reduces the risk of overfitting.”

Comment: I am a bit concerned regarding the composition of the FC metrics, given that they looked at time course similarity but from different perspectives. This should be carefully interpreted and validated. For example, classical correlation analysis, has the same behaviour of the proposed approach or is it specific to the set of metrics used?

Response: That is a very good point and one which we failed to properly justify in our manuscript. When planning which FC we wished to employ, we considered the possibility of using classical correlations. However, it has recently been shown that FC is more comprehensively captured by a combination of metrics that measure a range of different properties of the signal (e.g., linearity, non-linearity, time-domain, frequency-domain) and provide different kinds of information about brain interactions. In particular, Pearson correlations, which only capture linear relationships between brain signals, performs poorly in the presence of non-linearities. This was shown

empirically by Mohanty et al. (2017) who, despite observing a near-0 Pearson correlation between brain regions, detected high FC using alternate metrics (e.g., Wavelet Coherence).

Thus, Mohanty et al.'s work strongly indicates that using Pearson correlations alone might miss important aspects of FC, especially when looking at individual differences as we do in the present paper. Thus, we use a composite metric based on the nine metrics recommended by Mohanty and colleagues. However, we acknowledge that it is possible (and even likely) that our FC composite metric (and that of Mohanty et al. by extension) still did not fully capture the entire FC dynamics of our signals.

Nevertheless, we believe that the composite metric still reflects more accurately FC than having used any single metric alone, as Mohanty et al. observed in their study.

We also ran our entire pipeline using Pearson correlations as the sole metric of FC. With respect to PL (FL did not show any significant connection), four out of the five connections identified using the composite metric were also significant when using Pearson correlations alone (IAMY-IPFC, ICEB-rPFC, IHIP-IACC and rAMY-rACC).

Furthermore, we extracted the estimates from the Ridge regression model using the composite metric as FC, as well as those from the Ridge regression model using Pearson correlations alone. The correlation between these estimates was highly significant ($r = .60$, $p < .001$; new Fig. S38; see below), indicating that the estimates from Pearson correlations go in the same direction as those observed for the composite metric.

Figure showing the correlation of the Ridge regression estimates between the model that used the composite metric as FC metric and the model which used Pearson correlations.

We have revised the methods section where we describe FC, in which we added a new paragraph at the end mentioning these issues. The entire section now reads:

“Recently Mohanty et al.⁷⁹ tested the accuracy of several FC metrics by evaluating a support vector machine classifier using a neighbourhood component analysis feature selection. They found that FC was better characterised by a combination of nine different but complementary metrics (a composite metric) than any metric alone. The authors reasoned that Pearson correlations - the most common measure of FC - only look for statistical linear time-dependencies, whereas there could still be underlying statistical dependencies between BOLD signals that are poorly captured by Pearson correlations (e.g., non-linear dependencies).

Therefore, for the present study, we followed Mohanty's recommendation and computed a total of nine functional connectivity metrics: Pearson correlation, cross-correlation, dynamic time warping, Euclidean distance, Manhattan distance, Wasserstein distance, mutual information, coherence and wavelet coherence (see Supplemental Methods, Table S4 for the mathematical implementation, Fig. S7 for correlations among these metrics, and Fig. S8 for comparisons among the different FC metrics within our network).

Note that we also repeated all analyses using Pearson correlations as the FC metric in all our models. Results were largely consistent, with very similar connections in the analyses using the composite metric and Pearson correlations. In addition, the regression estimates from the two approaches were strongly correlated ($r = .60$, $p < .001$; Fig. S38), indicating that our findings are not critically dependent on the choice of FC metric. Nevertheless, while the composite metric may provide a broader coverage of FC, we acknowledge that even this metric may not fully capture the complexity of FC dynamics, and future work will be needed to refine these metrics further."

Comment: Renewal is included in two studies. Do the differences of extinction and acquisition hold for two studies as well?

Response: Thank you for this insightful comment. We agree that verifying whether the effects observed for acquisition and extinction also hold when restricting the analysis to the renewal studies is very important and relevant.

When we re-ran the LASSO models including only the two renewal studies (S2 and S4), all seven SC connections identified in the entire sample (IAMY–IACC, IAMY–IHIP, ICEB–rACC, IHIP–IPFC, rAMY–rACC, rHIP–rACC) were also detected.

Similarly, six of the seven FC connections identified in the entire sample (ICEB–rPFC, IHIP–IACC, IHIP–IPFC, rACC–rPFC, rCEB–IHIP, rHIP–rACC) were reproduced when analysing only the renewal studies.

These results indicate that the models are highly consistent, regardless of whether they are estimated using the entire dataset or only the renewal studies.

Comment: You delivered an electrical stimulus to the skin. Since I am not an expert of this type of stimulation. Are there any possible confounds in the brain responses of this stimulation? If yes, how did you control for these confounds?

Response: That is a reasonable concern. However, we should note that we analysed FC from resting-state fMRI and SC from diffusion MRI without any experimental task, so there should be no confounds of electrical/visceral stimulation in our brain responses. The SCR data we used for the dependent variable in our models came from the experiments conducted after the resting-state and diffusion measurements.

But, in general, most previous studies analysed the portion of the timeseries that correspond to the events before the electrical stimulation, so that the brain responses reflect the anticipatory response to the US rather than the stimulation per se (e.g., by defining separate CS-related and US-related regressors in a GLM).

Comment: The renewal behavioural results look alike. I mean that I would expect that CS+ has an increase rather than a decrease. What am I missing?

Response: That is correct. During renewal, we expect the CS+ to show an overall increase relative to CS-, which is what we found at the beginning of the renewal phase. Fig 3A, however, shows the time-course of renewal (i.e., across trials). Renewal is always tested in the context of acquisition, so we should expect the initial renewal trials to show an increase in the CR relative to late extinction (see left figure below). However, as with extinction, this process is followed by a gradual return of the CR to baseline levels, as participants realise that there is no US and that the CS is safe now, even in the acquisition context.

We have added a figure to the supplements (Fig. S39, also see left figure below) that may help visualise this difference. As can be seen below, the amplitudes for CS+ dropped to baseline levels during the late extinction phase (non-significant difference between CS+ and CS-, $p = .95$), whereas they increased substantially during the renewal phase compared to both late extinction (significant difference in CS+ between late extinction and renewal, $p < .001$) and CS- trials (significant difference between CS+ and CS- during renewal, $p < .001$). Importantly, there was no difference between late extinction and renewal for CS- trials ($p = .60$), indicating that the increase in amplitude for CS+ (renewal) trials cannot be accounted for by a general increase in the SCRs during the renewal phase (interaction $p < .001$)

One could argue that we do not need to analyse renewal on a trial-by-trial basis, and, instead, we could have taken the first trial (or mean) amplitude difference between CS+ and CS-. That is, indeed, a fair point. However, our decision to use trial-by-trial learning instead of fixed averages was because we wished to include a similar type of dependent variable for all experimental phases. Both acquisition and extinction were modelled from the trial-by-trial amplitudes of each respective phase. We believe our procedure is not only more informative than simply computing averages (which discards trial-by-trial dynamics), but also more robust given that using the initial renewal trials would likely be unstable.

We have now tested how similar our learning measure is to the initial renewal trial and there was a strong correlation ($r = 0.71$, $p < .001$) between the two measures, thus indicating that our learning measure (across trials) is very strongly related to the magnitude of initial renewal (see right figure below; both figures have been added to Fig. S39 in the Supplements).

Left: Compared to late extinction, renewal showed a return of the conditioned response for CS+ trials alone. **Right:** The estimates for the initial renewal trials (first CS+US-) strongly correlated with the renewal learning measure across trials.

Comment: The choice of the ROIs is an a-priori choice, motivated by literature, but to validate the findings the authors should test whether and how this choice is specific to this network or can be found with other ROIs.

Response: Thank you for this comment. To assess how our model behaves in other resting-state networks (RSNs), we ran our entire processing pipeline on several canonical RSNs. These RSNs were identified by Schaeffer et al. (2018), who used a gradient-weighted Markov Random Field model on resting-state data from 1,489 subjects to generate functionally homogeneous cortical parcels that align with known anatomical and functional boundaries. We used a parcellation with a 100-parcel resolution that included six canonical RSNs that show some overlap with the regions of interest in the fear and extinction network (Default mode network [DMN], dorsal attentional network [DAN], salience network [SN], frontoparietal control network [FPN], somatomotor network [SMN], and limbic network [LN]; see figure below). The Schaeffer2018 atlases were projected onto the functional space of each subject, and resting-state activity was averaged across each ROI of the networks. Functional connectivity was computed in the same way as for the original fear and extinction network (EXN). Overall explained variance was used to compare performance among the different RSNs.

As can be seen in the figure below, our EXN explained more variance than any other RSN. However, the LIN and DMN had the next highest explained variance compared to the remaining RSNs for PL and All groupings, respectively. Interestingly, LIN and DMN were also the RSNs that showed the highest spatial overlap with our EXN (see table below), suggesting that the predictive power of the RSN increases as the RSN becomes more similar to the EXN. Note that we excluded FL, since our analyses showed that FC did not significantly predict learning in this subset of studies when considered separately.

These results are shown in supplemental Fig. S29.

Top: Six resting-state networks (Schaeffer et al., 2018) used in this analysis. **Bottom left:** Explained variance for each of the RSNs after re-running our entire FC pipeline. **Bottom right:** Mean overlap between our fear and extinction network (EXN) and the other RSNs. DMN=Default Mode Network (pink); EXN=Extinction Network; FPN=Frontoparietal Network (dark blue); LIN=Limbic Network (light blue); SAN=Saliience Network (green); SMN=Somatomotor Network (red); DAN=Dorsal Attention Network (yellow).

Comment: In Figure 3B, results are a bit counterintuitive. I mean that during FL you found no important connections while when including S4, FC connectivity results to be important. However, when using S4 alone FC is still important. This looks weird or at least should be discussed and analyzed more in depth to understand how this has happened.

Response: Again, we thank the reviewer for raising this point. While we agree that this may appear surprising, it is entirely consistent with known properties of regularised regression. In particular, LASSO selects predictors under a penalty and, therefore, it is sensitive to the consistency of a predictor's effect across the training data and to the correlation structure among predictors.

In our data, PL consists of very similar paradigms and shows a consistent relationship for several connections. In contrast, the FL group consists of more heterogenous paradigms (e.g., with different stimuli, trial numbers, and US types) and, therefore, shows much larger between-study variability. This is corroborated by an overall much larger explained-to-unexplained variance ratio for PL (.17) than FL (.006).

As a result, predictors that are predictive in PL alone may be more unstable in FL and are, consequently, shrunk to zero. When PL and FL are combined, the consistent PL signal is sufficiently strong to be selected in the pooled model. However, we would like to mention that for the majority of connections, FL studies still did show an effect in the same direction as PL, but with slightly larger variability or decreased slope, as shown in the figure below:

Regression models showing the relevant FC connections identified using the entire sample (see Fig. 3B, FC panel). Note that both FL and PL show a similar directionality in the regression slopes for all connections.

We added a new paragraph at the end of the “Limitations of the present study” section where we discuss this point:

“All experiments analysed here fell under either the FL or the PL paradigm. However, FL studies consisted of relatively heterogeneous experiments (e.g., with different stimuli, trial numbers, and US types), whereas PL studies were more homogenous. On the other hand, the heterogeneity of FL studies also allowed us to find common predictors that generalised across a broad range of experimental conditions, which would not have been possible when only a single paradigm had been conducted. In this vein, future work may aim to include an even greater variety of paradigms, such as eye blink conditioning, appetitive, and olfactory learning.”

Comment: The authors mentioned that “Note that it is possible for a connection to be identified as significant when data are pooled across all studies, even it was not significant in any individual study”. Can you elaborate on this? I mean this is crucial for establishing the validity of your results, can you provide a clear example or simulation that validates your sentence.

Response: Thank you, this is an important point related to the previous comment. That sentence is referring to the typical observation that predictors in regression models may be significant when data are pooled across many groups but not significant when data are split into smaller subsets.

Specifically, if every study has a small positive/negative true effect, but they also have some noise, then the estimates for each study alone can turn out non-significant. Pooling across studies increases the total sample size, which reduces the standard errors of the estimated coefficients (the standard errors are inversely related

to the square root of the total sample size). Consequently, the pooled estimator can become statistically significant even though no individual study has enough power to detect the small effect. This is not unusual or paradoxical; it is simply the result of pooling weak (but consistent) signals.

We also include a short simulation (see https://github.com/caadgomes/extinction-learning-study/blob/main/scripts/simulation_regression.R) showing that a small but consistent within-study betas can be non-significant in individual groups but become significant when pooled across groups.

Comment: In Figure 4J-L, it is not clear what the colors are indicating.

Response: We apologise for this oversight. The blue and red colours represent the PL and FL paradigms, respectively. We have now added the legend to Fig. 4J-L.

Comment: How did you find that some EC connections are inhibitory and others are excitatory? I mean, as far as I know, EC gives you the direction of the connections so if a region is influenced or influences another. While the excitatory and inhibitory mechanisms are more an interpretation of the directionality. I would suggest reframing the discussion to assess for the intrinsic limitation of DCM analyses.

Response: The reviewer is completely correct. DCM provides estimates indicating if the influence of one region on another is positive or negative, which we interpreted as being “excitatory” and “inhibitory”, respectively. While we followed the terminology typically used in DCM studies (see for example Razi et al., 2017; Friston et al., 2014; Almgren et al., 2020), as the reviewer correctly points out, these labels reflect the sign of the inferred causal effect, not the actual measurement of the underlying synaptic mechanisms.

We have included a paragraph in the Methods section on page 29 pointing the reader to the limitation identified by the reviewer, while still using the labels “excitatory” and “inhibitory” in the main text to be consistent with the terminology used in other DCM-related papers:

“Note that we use the terms “excitatory” and “inhibitory” in the context of EC to denote influences that increased (positive sign) or decreased (negative sign) activity in target regions, respectively. Even though the excitatory/inhibitory labelling in the present study does not refer to cellular or synaptic mechanisms – as we have not directly measured neurotransmitter action – we, nevertheless, use these labels in the text to be consistent with the terminology used in other DCM-fMRI studies³¹”.

Comment: I would recommend to add the versions of software used in a requirements.txt or project.toml file.

Response: We now include a requirements.txt file indicating the versions of all packages, libraries and software used in the present study.

References

Almgren, H. et al. (2020). The effect of global signal regression on DCM estimates of noise and effective connectivity from resting state fMRI. *Neuroimage*.

Friedrich, S. et al. (2023). Regularization approaches in clinical biostatistics: A review of methods and their applications. *Stat Methods Med Res*.

Friston, K. et al. (2014). A DCM for resting state fMRI. *Neuroimage*.

Hastie, T., Tibshirani, R., & Friedman, J. (2009). The Elements of Statistical Learning: Data Mining, Inference, and Prediction. *Springer*.

Tibshirani, R. (1996). Regression shrinkage and selection via the lasso. *Journal of the Royal Statistical Society*.

Razi, A. et al. (2017). Large-scale DCMs for resting-state fMRI. *Netw Neurosci*.